# Modelling long-term blanket peatland development in eastern Scotland.

Ward Swinnen[1,2], Nils Broothaerts[1], Gert Verstraeten[1]

[1]Department of Earth and Environmental Science, KU Leuven, Leuven, 3000, Belgium

[2]Research Foundation Flanders - FWO, Brussels, 1000, Belgium

*Correspondence to*: Ward Swinnen (ward.swinnen@kuleuven.be)

**Abstract.** Blanket peatlands constitute a rare ecosystem on a global scale but is the most important peatland type on the British Isles. Most long-term peatland development models have focussed on peat bogs and high-latitude regions. Here, we present a process-based 2D hillslope model to simulate long-term blanket peatland development along complex hillslope topographies.

To calibrate the model, the peatland architecture was assessed along 56 hillslope transects in the headwaters of the river Dee (633 km²) in eastern Scotland, resulting in a dataset of 866 soil profile descriptions. The application of the calibrated model using local pollen-based land cover and regional climate reconstructions (mean annual temperature and mean monthly precipitation) over the last 12,000 years shows that the early-Holocene peatland development is largely driven by a temperature increase. An increase in woodland cover only has a slight positive effect on the peat growth potential contradicting the

hypothesis that blanket peatland developed as a response to deforestation. Both the hillslope measurements and the model simulations demonstrate that the blanket peatland cover in the study area is highly variable both in extent and peat thickness stressing the need for spatially distributed peatland modelling. At the landscape scale, blanket peatlands were an important atmospheric carbon sink during the period 9.5 ka – 6 ka BP. However, during the last six thousand years, the blanket peatlands are in a state of dynamic equilibrium with minor changes in the carbon balance.

## 1 Introduction

Peatlands occur across the globe and contain up to one third of the global soil carbon stock, despite covering approximately less than three percent of the Earth's surface (Gorham, 1991; Xu et al., 2018). Especially at higher latitudes, peatlands are an important ecosystem type and their dynamics have profoundly influenced the terrestrial carbon cycle throughout the Holocene (Yu et al., 2011). Unfortunately, little is known about long-term peatland dynamics and their response to climatic and land

cover changes (Wu, 2012).

Blanket peatlands are spreads of peat of varying thickness, covering the underlying topography, thus "blanketing" the landscape (Lindsay, 1995). This peatland type occurs in hyperoceanic climates with cool and moist conditions throughout the year, and is mostly confined to the maritime edges of the continents (Gallego-sala and Prentice, 2013). Due to their location in the landscape, blanket peatland formation is more controlled by topography, compared to other peatland types (Parry et al.,

2012). Although rare on a global scale, up to 6 percent of the area of the United Kingdom is covered by blanket peatland (Jones et al., 2003). The large extent of the Scottish blanket peatlands, covering 23 percent of the land area, compared to the international rarity of these environments, make the Scottish peatlands of high conservation value (Fyfe et al., 2013; Tipping, 2008).

During the Holocene period, large areas of blanket peatland have developed throughout the Scottish Highlands and this shift
from mineral to waterlogged and nutrient-poor organic soils is one of the most important Holocene landscape changes in Scotland. Different hypotheses have been raised regarding the cause of this peatland development (Tipping, 2008). The original hypothesis, as proposed by Moore, linked the blanket peatland initiation to human impact, where anthropogenic land use change and increased grazing during the Neolithic period led to a shift in the hillslope hydrology resulting in the paludification of the upland soils (Moore, 1973). While this hypothesis has been supported by local studies throughout the British Isles, other
authors have suggested that, at least for Scotland, the initiation of blanket peatlands resulted from climatic changes during the Atlantic period (Ellis and Tallis, 2000; Huang, 2002; Simmons and Innes, 1988; Tipping, 2008). A recent study based on a database of basal radiocarbon dates shows regional differences in the timing of the blanket peatland development with an earlier timing for Central and Southern Scotland, compared to the other regions of the British Isles (Gallego-Sala et al., 2016). Most of the case studies studying the blanket peatland initiation are based on field data such as pollen cores and radiocarbon
dating, but studying causalities based on timing alone is difficult (Gallego-Sala et al., 2016). Process-based modelling of this landscape transformation could prove to be a useful technique, complementary to the field data, to provide insight in the underlying processes and mechanisms.

In recent decades, several peatland models have been developed, varying in spatial and temporal scale and in model complexity (Frolking et al., 2010). A good overview of the models developed for simulating long-term peatland behaviour is given by
Baird et al. (2012). Currently, several long-term peatland models such as Digibog and the Holocene Peatland Model (HPM) allow to simulate peatland processes and the feedbacks between ecology, hydrology and peat properties over Holocene timescales (Baird et al., 2012; Frolking et al., 2010). These models have been applied successfully within the context of peat bogs, but are difficult to transfer to blanket peatlands for two reasons. Firstly, these models are developed as cohort models, where each year, a new peat layer is added to the soil profile and included in the calculations for the remaining part of the
simulations. As a result, these allow to simulate temporal changes in peat properties such as hydraulic conductivity within the peat profile, but as the simulated time period increases, these models become computationally expensive, especcially when a spatial dimension is added. Secondly, these models have been developed for peat bogs, which have a different peatland architecture compared to blanket peatlands and are therefore not always adapted to simulating peatland processes along complex hillslope topographies. These issues have been partially resolved by the MILLENNIA model, which has been
designed specifically for blanket peatlands (Heinemeyer et al., 2010). This model is also a cohort model but incorporates additional processes which are specific to blanket peat such as runoff-driven peat erosion. However, the high degree of detail in the model domain and the representation of the processes make it difficult to apply cohort models at the landscape scale.

In this study, a process-based peatland model is presented which is able to simulate the hillslope hydrology and peatland dynamics along topographically complex hillslopes on Holocene timescales. Additionally, the representation of the model domain is relatively simple using a diplotelmic peat profile, making it computationally feasible to study peatland development on a landscape scale by simulating a large number of hillslope cross-sections. The model is applied to the Upper Dee catchment in the Cairngoms National Park, eastern Scotland. The goal of this study is twofold. Firstly to apply a relatively simple process-based peatland model to study the long-term blanket peatland development in the Scottish Highlands on a landscape scale. Secondly, to identify the relative importance of climatic and land cover changes in the long-term blanket peatland development.

## 2 Materials and methods

### 2.1 Study area

The study area consists of the headwaters of the river Dee in eastern Scotland, with an elevation ranging from 322 to 1309 metres a.s.l. The area lies in the centre of the Cairngorms National Park and is managed by the Mar Lodge, Invercauld and Mar estates (fig.1). The geology of the Dee catchment is characterised by metamorphic and igneous rocks, with schists and granulites in the southern part of the study area and granite batholith intrusions in the north (Maizels, 1985). The entire area was glaciated by the Scottish ice sheet during the last ice age, which retreated between approximately 16 ka and 13.6 ka BP. In contrast to the Western Highlands, the Cairngorms massif was not subjected to widespread glacial expansion during the Younger Dryas (Loch Lomond Stadial). During this period, the glacial activity remained largely restricted to the cirques (Everest and Kubik, 2006). The development of the current landscape and soils in the upper Dee area has been influenced by the deglaciation, forming a wide variety of glacial and fluvioglacial landforms (Ballantyne, 2008). In many parts of the study area, the bedrock is covered by glacial till of varying thickness (Maizels, 1985). The summits and ridges mostly carry skeletal soils and bedrock outcrops, while the slopes are covered by blanket peat and alpine podzols (Smith, 1985). The peat deposits are found both lying directly on bedrock and overlying a layer of mineral sediment. This mineral substrate consists of gravel-rich silt loam and sandy loam in the southern part of the study area and sandy loam to loamy sand in the northern part.

Currently, the area is dominated by semi-natural land cover, including alpine and montane heath vegetation on the highest summits, heather moorland and small pockets of natural forest (Tetzlaff and Soulsby, 2008). The total annual precipitation ranges from 800 mm in the eastern part of the study area to almost 2000 mm on the mountain tops, with a significant proportion of the precipitation falling as snow during the winter months (Dunn et al., 2001). The temperature regimes can vary considerably within the study area. The town of Braemar (339 m a.s.l.), has a mean annual temperature of 6.8 °C, ranging from 1.6 °C as a mean winter temperature (DJF) to 12.8 °C as a mean summer temperature (JJA). In contrast, the summit of Cairn Gorm (1245 m a.s.l.) has a mean annual temperature of 0.6 °C, and ranges from a mean winter temperature (DJF) of -2.6°C to a mean summer temperature (JJA) of 5.3 °C.

Early Holocene traces of human presence have been found within the study area, with archaeological evidence indicating the presence of Mesolithic hunter-gatherer structures as early as 8.2 ka cal BP in the western part of the study area (Warren et al.,

2018). The first traces of permanent settlement in the Upper Dee are from the village of Braemar around 1000 AD (Paterson, 2011). In contrast to the western part of Scotland, the study area shows no traces of large scale peat extraction, which is probably due to the relatively thin peat profiles and difficult access to the area (Maurer, 2015).

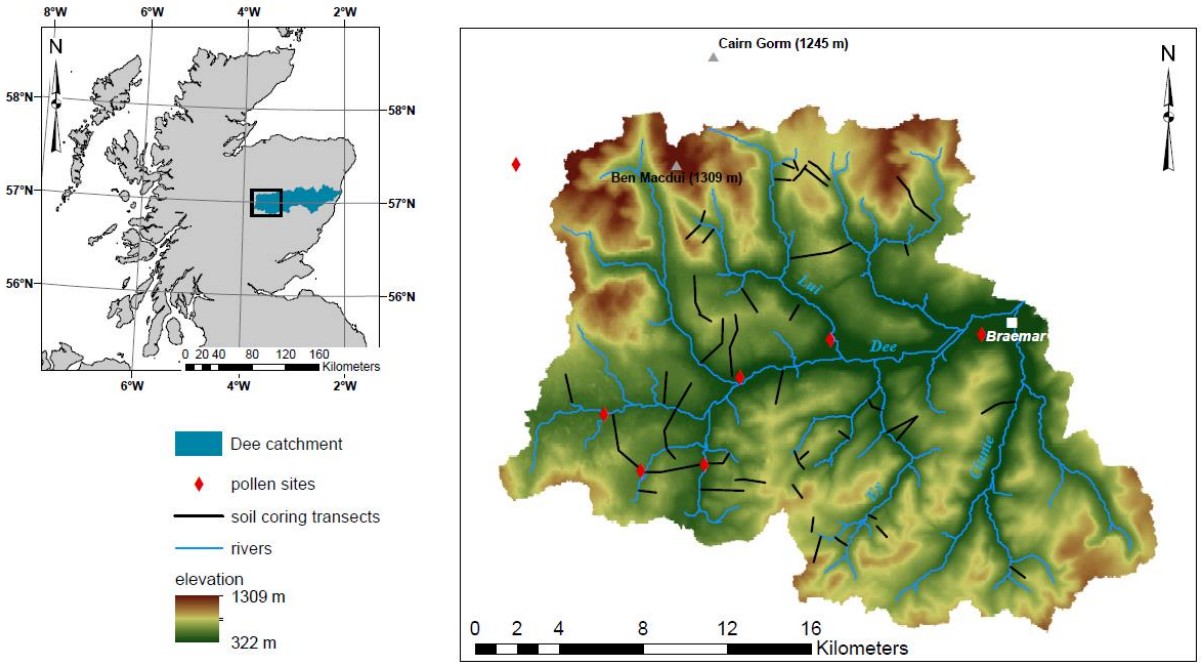

**Figure 1: Location of the Upper Dee catchment, with indication of the hillslope transects and the pollen sites used for the land cover**
**reconstruction.**

### 2.2 Field data

For the study area, the blanket peatland architecture was assessed along 56 hillslope transects across the study area during field campaigns in 2015 and 2017 using soil corings, laboratory analysis of peat samples and pollen cores (fig. 1). The soil corings were taken along the hillslope transects with a spacing of approximately 50 metres using a gauge auger. The hillslope
topography was measured using Post-Processing RTK GPS measurements. The transect locations were selected in order to include a wide variety of lithologies, elevation zones and topographic parameters such as aspect, slope and curvature. Additionally, the carbon content, dry bulk density and water content of the peat deposits were derived from 35 field samples, collected as core sections with a length of 5 centimetres. These samples were collected at random coring locations and at depths ranging from 5 centimetres to 165 centimetres below the surface. The regional vegetation evolution over the past 12,000
years was reconstructed based on seven pollen cores located within the study area (fig. 1). These cores provide vegetation information from different elevation zones and with a varying distance to the low-lying valleys of the Dee and the Spey (Birks, 1969; Hunter, 2016; Huntley, 1994; Paterson, 2011). Using the REVEALS model (Regional Estimates of Vegetation Abundance from Large Sites (Sugita, 2007)), the pollen percentages were converted to regional vegetation fractions. REVEALS was developed to reconstruct regional vegetation composition using pollen data from large lakes, but previous

studies has shown that a group of sites can also be used to estimate regional vegetation cover (Fyfe et al., 2013; Mazier et al., 2012; Trondman et al., 2016). Pollen type parameters (pollen productivity and fall speed) were based on the standardized set of Mazier et al., (2012). The regional vegetation fractions were grouped in five classes (coniferous trees, deciduous trees, shrubs, heather, grasses & herbs) and used as land cover input in the hillslope model. As the land cover reconstruction for Scotland based on REVEALS by Fyfe et al. does not include pollen data from high-elevation sites, a new land cover reconstruction was made by Hunter (2016) for this study using local pollen data (Fyfe et al., 2013; Hunter, 2016).

At all coring locations, the soil profiles were described based on visual inspection, analysing the colour, texture and possible presence of macroscopic remains (charcoal, wood, …). Based on the coring descriptions, the peat thickness could be derived. In this study, peat is defined as a dark organic-rich layer of minimal 10 centimetres thick without or with minimal presence of mineral material based on visual inspection. Organic-rich horizons with a clear presence of mineral material were not classified as peat.

In total, 34 peat samples were radiocarbon dated at 17 locations throughout the study area, encompassing a range of topographic situations and peat thicknesses (see appendix A2 for dating details). All radiocarbon dates were performed by the Belgian Royal Institute for Cultural Heritage and calibrated using the Oxcal 4.3 software and the IntCal13 calibration curve (Bronk Ramsey, 2009; Reimer et al., 2013).

## 2.3 Model outline

The model presented here is based on the concept of impeded drainage, where environmental conditions such as bedrock topography can cause waterlogging and peat formation (Alexandrov et al., 2016; Clymo, 1984; Ingram, 1982). The basic structure of the hillslope peatland model consists of a hydrology module simulating the water table behaviour along the hillslope, which is coupled to a peat growth module simulating biomass production and decomposition (fig. 2). The model domain consists of a two-dimensional hillslope cross-section, which is discretised in a series of model gridpoints. The stratigraphy consists of an impermeable bedrock, overlain by a layer of glacial till, which is assumed to be porous. Over time, a peat profile can develop on top of this till substrate when the right environmental conditions are met. The hillslope topography is based on the detailed GPS-measurements for each coring location.

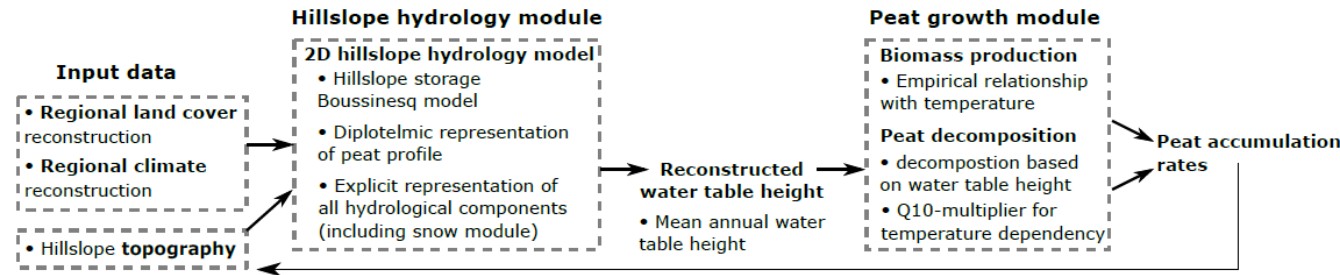

**Figure 2: General model workflow. For a more detailed description of the model structure, the reader is referred to the text.**

### 2.3.1 Hillslope hydrology module

The water table dynamics are modelled using a variant on the Boussinesq equation for a non-constant slope (Hilberts et al., 2004):

Eq. (1)  $\varepsilon \frac{\partial S}{\partial t} = \frac{k}{\varepsilon} \cos i(x) \left[ B \frac{\partial S}{\partial x} + S \frac{\partial B}{\partial x} + \varepsilon S \frac{\partial i(x)}{\partial x} \right] + \frac{k}{\varepsilon} \sin i(x) \left[ \varepsilon \frac{\partial S}{\partial x} - SB \frac{\partial i(x)}{\partial x} \right] + \varepsilon N$

With  $B = \frac{\partial}{\partial x} S$

With $x$ the distance to the hillslope bottom (m), $\varepsilon$ the soil porosity (m m$^{-1}$), $S$ the actual water storage (m), $k$ the hydraulic conductivity (m s$^{-1}$), $i$ the bedrock slope (m m$^{-1}$) and $N$ the rainfall recharge or infiltration (m) (Hilberts et al., 2004). The Boussinesq equation is a simplified form of the full Richards's equation for unsaturated soils by excluding a representation of the unsaturated zone. This simplification can be justified for peat soils given the often shallow position of the water table in peatlands (Ballard et al., 2011; Paniconi et al., 2003). As a result, the Boussinesq equation assumes an instantaneous exchange of water (e.g. infiltration and evapotranspiration) between the surface and the saturated zone of the soil profile. This simplified representation leads to a significant reduction of the computational time (Ballard et al., 2011). To enable the use of the Boussinesq-equation for the simulation of the hillslope hydrology, local topographic depressions are filtered out. In this study, the Boussinesq equation is discretised using a forward in time – centred in space finite difference scheme for the diffusion component and a first-order upwind finite difference scheme for the advection component (Campforts and Govers, 2015).

The diplotelmic nature of the model is represented by the depth-integrated saturated hydraulic conductivity. Each stratigraphic unit (mineral substrate, catotelm and acrotelm) has a specific saturated hydraulic conductivity value. The bedrock is excluded from the water table depth calculations as it is assumed to be impermeable.

A simple snow module is included in the hydrological model, with precipitation falling as snow during periods with sub-zero temperatures. The amount of melt is based on a degree day factor model. Snow sublimation is not explicitly represented in the model. For each timestep, infiltration- and saturation excess overland flow is calculated. The produced runoff is assumed to leave the hillslope before the next timestep. For open peatland vegetation types, all rainfall is assumed to be able to infiltrate for intensities below 2 mm h$^{-1}$. For higher intensities, the infiltration rate increases with higher precipitation rates:

Eq. (2)  $ir = 0.626 * p + 0.0002$

With $ir$ the infiltration rate (mm h$^{-1}$), $p$ the precipitation rate (mm h$^{-1}$) (Holden and Burt, 2002). For woodland peatlands, infiltration rates of up to 30 mm h$^{-1}$ are reported (Cairns et al., 1978). In the model, this maximal infiltration rate of 30 mm h$^{-1}$ is used for a fully forested peatland. The final infiltration rate at a certain location is determined based on linear interpolation between the infiltration rates of open and forested peatland based on the percentage woodland cover at each model gridpoint.

The potential plant transpiration and soil evaporation (mm day$^{-1}$) are calculated separately based on the Leaf Area Index (*LAI*), which enables differentiation based on the vegetation cover (eq. (3)) (Williams et al., 1983).

Eq. (3)  $E_{soil} = E_{pot} e^{(-0.4 LAI)}$

$E_{plant} = \frac{E_{pot} LAI}{3}, 0 \leq LAI \leq 3$

$$E_{plant} = E_{pot} - E_{soil}, LAI > 3$$

With $E_{soil}$ the soil evaporation rate (mm day$^{-1}$), $E_{plant}$ the plant transpiration rate (mm day$^{-1}$) and $E_{pot}$ the potential evapotranspiration rate (mm day$^{-1}$), which is calculated using the Thornthwaite equation based on the mean monthly temperature.

The actual evapotranspiration rate (*AET*) (mm day$^{-1}$) is calculated as a function of the water table depth (eq. (4)). If a gridpoint consists of glacial till without a peat cover, the *AET* is at the potential rate when the water table is at the surface ($z_1 = 0$) and

180 decreases linear until depth $z_2$. If peat is present, the actual evapotranspiration is assumed to be at the potential rate if the water table is located in the upper horizon ($z_1 =$ acrotelm thickness) and decreases linearly until a depth $z_2$ (m) (Frolking et al., 2010; Lafleur et al., 2005). In this study, $z_2$ is set to 1 meter for both the glacial till and peat soils. In contrast to more detailed peatland models such as the MILLENNIA model, the AET – water table depth relationship is not influenced by changes in vegetation groups and their root properties, resulting in constant values for $z_1$ and $z_2$ throughout the simulations (Carroll et al.,

2015).

Eq. (4)   $AET_t = \left( E_{soil} + E_{plant} \right) \frac{z_2 - wt}{z_2 - z_1}, for\ z_1 \leq wt \leq z_2$

Since detailed local climate reconstructions are scarce, several peatland modelling studies have used continental or at best regional climate reconstructions (mostly pollen-based) which were fine-tuned using local climate information (Frolking et al., 2010; Heinemeyer et al., 2010; Morris et al., 2015). Here, a similar approach is used. Input data for temperature and

190 precipitation values are based on a European gridded dataset of mean annual temperature and mean monthly precipitation anomalies for the last 12,000 years derived from pollen data with a spatial resolution of 1°x1° and a temporal resolution of 500 years (fig. 3) (Mauri et al., 2015). As total annual precipitation amounts and mean annual temperatures vary considerably throughout the study area, the precipitation and temperature data were corrected for orographic effects. Data from eight meteorological stations in the vicinity of the study area were used to construct linear regression models, correcting the mean

daily precipitation amount and mean annual temperature for each location based on the elevation (eq. (5) – eq. (6)) (see appendix A1 for the weather station details):

Eq. (5)   $P = (0.003776 * E) + 1.669$

Eq. (6)   $MAT = (-0.0083 * E) + 13.2839$

With *P* the mean daily precipitation amount (mm), *E* the elevation (m a.s.l.) and *MAT* the mean annual temperature (°C).

Additional local climate information was added to the climate reconstruction data by incorporating random variability to the reconstructed temperature and precipitation data, based on the variability as observed in the weather station of Braemar for the period 1853 – 2010. The relatively low spatial and temporal resolution of the continental-scale climate reconstruction will probably lead to the underrepresentation of short-lived events and local climate variability. However, by incorporating the orographic corrections and random variability, local climate information is used to fine-tune the records and increase both the

spatial and temporal variability. The time series used as model input are based on daily temperature and hourly precipitation data from the weather station of Braemar, which are rescaled using both the regression equations for elevation effects and the

long-term anomalies for temperature and precipitation (fig. 3). As a result, precipitation and temperature series with a high temporal resolution are used throughout the entire studied period. The model is run with a spatial resolution of 50 metres, similar to the average coring distance. The time resolution is set to 400 seconds for the hillslope hydrology module and 1 year for the peat growth module to ensure model stability.

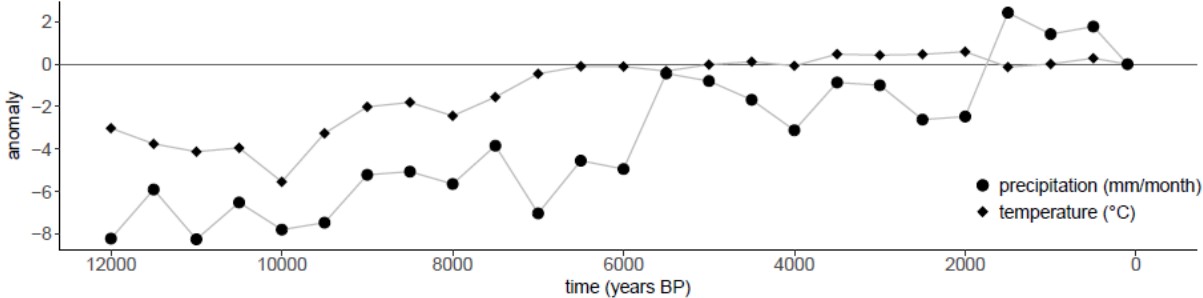

**Figure 3: Reconstructed mean annual temperature (°C) and mean monthly precipitation (mm month[-1]) anomalies for the period 12 ka – 100 BP with a 500-year interval for the location of Braemar. Values extracted from a gridded European dataset with a spatial resolution of 1x1 degree (Mauri et al., 2015).**

## 2.3.2 Peat growth module

The peat accumulation at each gridpoint is calculated as the balance between biomass production and decomposition. In the literature, several relatively simple equations can be found to calculate the Net Primary Production (NPP) based on climatic data (mean annual temperature, total annual rainfall or potential evapotranspiration) such as the Miami model or the Thornthwaite Memorial Model of which some have been implemented in peatland models (Heinemeyer et al., 2010; Lieth, 1973; Lieth and Box, 1972). In some cases, such as the Miami model, the NPP is calculated based on multiple climatic variables (rainfall and temperature), using the minimum value of both equations as the final NPP-value. However, given the climatic conditions of the Scottish Highlands, precipitation is not a limiting factor for biomass production. As a consequence, the biomass production is simulated as a function of the mean annual temperature using a power function regression equation based on field data for the Moor House Reserve in northern England (Garnett, 1998), corrected for the woodland cover. A possible disadvantage of this simple approach is the dependence of the biomass production calculations on the quality of the climate and land cover reconstructions.

Eq. (7) $\quad NPP = 60.06 \, (MAT^{1.134}) * (1 + (\frac{wc}{wc_{max}} * wi))$

With *NPP* the Net Primary Production (g m$^{-2}$ a$^{-1}$), *MAT* the mean annual temperature at the gridpoint location (°C), $wc_t$ the woodland fraction, $wc_{max}$ the woodland percentage of a fully forested peatland and *wi* the percentage increase in NPP between an open and wooded peatland. In general, the NPP is higher for wooded peatland compared to open peatland vegetation, with reported values of a 12 percent increase for bogs and 17 percent for fens (Beilman and Yu, 2001; Szumigalski and Bayley, 1997). In this study, *wi* is set to 15 percent and $wc_{max}$ to 40 percent.

The peat column at each gridpoint is divided in an oxic and anoxic zone based on the calculated mean annual water table height. The total decomposition can thus be written as:

Eq. (8)   $D = k_1 * wt + k_2 * (h - wt)$

With $D$ the total decomposition (m a$^{-1}$), $k_1$ and $k_2$ the rates of decomposition under anoxic and oxic conditions (yr$^{-1}$), $h$ the thickness of the soil profile above the bedrock (m) and $wt$ the height of the water table above the bedrock (m) (Hilbert et al., 2000). The decomposition rates are dependent on the mean annual air temperature using a $Q_{10}$-temperature multiplier, which is the ratio by which the biomass respiration rate increases under a 10 °C temperature increase. The range in $Q_{10}$-values mentioned in the literature is large, but Chapman and Thurlow demonstrated that $Q_{10}$-values are generally higher for temperatures between 0°C and 5°C (Chapman and Thurlow, 1998). This effect can be attributed to the fact that as temperatures rise above the freezing point, more microbial groups will become active, leading to relatively large changes in respiration rates for small changes in temperature. As a result, two $Q_{10}$-values are used in this study. A $Q_{10}$-value of 2.2 is used for temperatures above 5 °C, and 3.7 for temperatures between -4 and 5 °C. Below -4°C, the decomposition is assumed to cease completely (Chapman and Thurlow, 1998; Rosswall, 1973 as cited by Clymo, 1984; Wieder and Yavitt, 1994; Wu, 2012). The biological module runs at an annual timescale. Based on the calculated peat accumulation rate, the hillslope topography is updated annually.

### 2.3.3 Peatland initiation

Simulations start with a hillslope consisting of an impermeable bedrock covered by glacial till. As the thickness of the till is not known at each location, it is assumed to have a constant thickness of 50 centimetres. Over time, the organic matter accumulates within the upper 30 centimetres of the mineral soil forming an organic-rich horizon based on the balance between biomass production and decomposition. When a threshold is exceeded, additional organic matter which is produced, starts to accumulate as peat at that location, with the properties of an acrotelm. In this study, the threshold is set at an amount of organic matter, equivalent to a peat layer with a thickness of 10 centimetres using the median dry bulk density and organic carbon percentage of the 35 peat samples, collected in the field. This ensures that a similar definition for peat is used both for the hillslope corings as for the model simulations. Once the peat thickness exceeds the thickness of the acrotelm layer, the peat layer becomes diplotelmic, with the peat below the acrotelm having the properties of the catotelm. Once the biomass within the simulated peat profile decreases below the biomass threshold, the gridpoint is no longer considered to be covered by a peat layer and only mineral soil properties are taken into account.

### 2.3.4 Boundary conditions

The impermeable bedrock below the glacial till is used as a zero-flux boundary condition at the bottom of the model domain. At several locations throughout the study area, rivers have eroded the stream bank, exposing the peat. At the lower end of the hillslope, the water storage is thus set to a fixed value, representing the depth of the river. For the gridpoint at the top of the hillslope transect, a lateral zero-flux boundary is assumed.

## 2.4 Model calibration and validation

Model calibration is based on the measured mean peat thickness per topographic class. In total, nine topographic classes were defined by dividing both the measured slope and curvature at each coring location in three classes, resulting in nine possible combinations. The calibration procedure resulted in topographic class limits of 0.098 and 0.135 m m$^{-1}$ for slope and -0.184*10$^{-3}$ m$^{-1}$ and 0.184*10$^{-3}$ m$^{-1}$ for curvature. For all 56 hillslope transects, the modelled mean peat thickness per topographic class after 12,000 years of simulation is compared to the mean peat thickness measured in the field. In total, three model parameters were calibrated: the decomposition rates under oxic and anoxic conditions and the acrotelm thickness. The goodness-of-fit of each parameter combination was evaluated based on minimization of the Root Mean Squared Error (RMSE) between the mean modelled and measured peat thickness per topographic class. Out of the 866 hillslope corings, 433 were selected randomly to be used as calibration points and the others as validation points. Since the spacing between the soil corings is slightly variable, the model results were resampled to the locations of the soil corings using linear interpolation.

As an additional validation of the model behaviour, the simulated peat growth initiation dates for all model gridpoints can be evaluated against a dataset of basal radiocarbon dates for blanket peat deposits in the upland regions of Scotland, with an elevation above 300 metres a.s.l. (n = 30) (Gallego-Sala et al., 2016). The dataset was expanded by incorporating 17 additional basal radiocarbon dates on peat deposits from within the study area (see appendix A2 for dating details). For each of the 17 locations within the study area for which radiocarbon dates were available, the basal age was estimated using the clam 2.2 software package to construct age-depth models and extrapolate to the bottom of the peat layer (Blaauw, 2010). As the available initiation dates based on radiocarbon dating were not necessarily taken at the modelled transect locations, the comparison between modelled and observed peat growth initiation is based on the probability density curves using a bin width of 500 years. Depending on the amount of available radiocarbon dates at each location, some estimates of the peat growth initiation date were based on the extrapolation of the age-depth model over large sections of the peat profile. To analyse the effect of the age-depth model extrapolation on the resultant probability density curve, an additional probability density curve was constructed containing only those radiocarbon dated samples which were directly measured at the bottom of a peat layer (n = 20).

## 3 Results

### 3.1 Field measurements

In total, soil coring descriptions were made at 866 locations throughout the study area (detailed descriptions and location data can be found in the supplementary material). Based on the definition of peat used in this study, 57 percent of the coring locations contained a surface peat layer, with a mean measured peat thickness over all coring locations of 36 centimetres, with a maximal value of 3 metres. The mean measured peat thickness per hillslope transect varies between 0 and 96 centimetres (fig. 4). Overall, the transects with a high mean peat thickness can be found in the upstream parts of tributaries of the river

Dee. Strong spatial variability occurs, even at small distances, making the peat cover throughout the area highly variable in both occurrence and mean thickness. The mean measured peat thickness per topographic class ranges from 23.6 ± 31.9 cm for the class with a moderate slope and a convex curvature to 54.1 ± 65.3 cm for the topographic class with a low slope and a straight curvature. Based on 35 randomly selected soil samples, which were identified in the field as peat, the median organic carbon percentage was calculated at 51.9 ± 7.3 percent and the dry bulk density at 0.128 ± 0.063 g cm$^{-3}$.

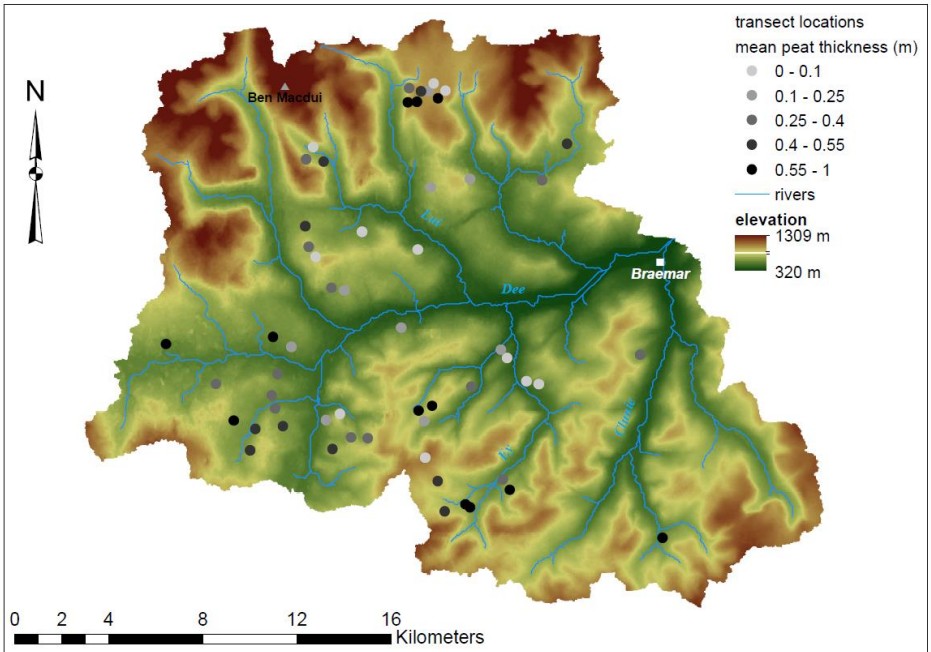

**Figure 4: Mean measured peat thickness per hillslope transects (n = 56).**

As the model is based on the principle of impeded drainage, there is an assumed relationship between the bedrock slope and the peat thickness at a certain location. The peat thickness data indicate that this relationship is present, showing a clear decrease in the maximum thickness with increasing bedrock slope. However, the variability, especially at lower slope angles indicates that shallow peat layers or even the absence of peat is observed for every slope value (fig. 5a). In the Boussinesq equation, the bedrock slope is used instead of the surface slope. One could argue that the bedrock slope might not relate directly to the surface slope in for example local depressions filled with peat. However, the comparison of the bedrock slope and surface slope for all coring locations indicate a clear and strong relationship between both, with the observed range in slope values strongly exceeding the differences between bedrock and surface slope at a single location (fig. 5b). As a result, the use of the bedrock slope is thus unlikely to introduce a bias in the modelled peat thickness values.

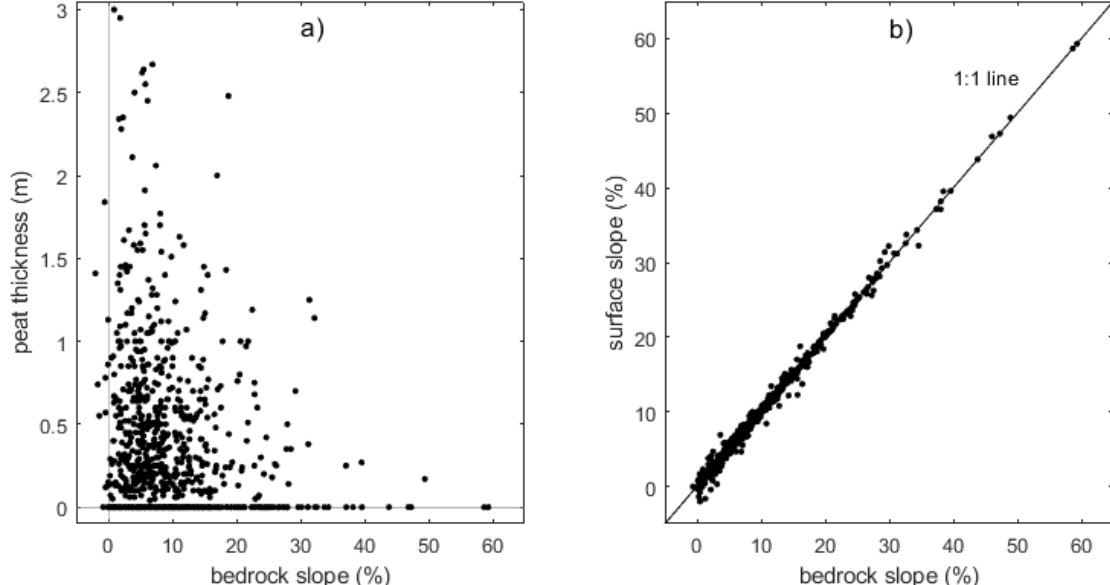

**Figure 5: a) Scatterplot of the measured peat thickness (m) as a function of the bedrock slope (%) and b) scatterplot of the surface slope (%) as a function of the bedrock slope (%) for all coring locations (n = 866). The slope calculations are based on the measured coordinates of the coring locations.**


The pollen-based reconstructed land cover shows an early-Holocene woodland increase until the period 8.4 ka - 7.2 ka cal BP (fig. 6). This period is followed by a general woodland decline, with the woodland cover dropping below 5 percent from 3.6 ka cal BP onwards. The reconstructions for the individual pollen cores show an important east-west gradient in terms of maximal forest cover, with higher woodland percentages for the eastern and lower-lying part of the study area (Paterson, 2011).

The woodland is of a mixed type containing both coniferous species (Scots pine) and deciduous species (birch, rowan and aspen). A study by Fyfe et al., reconstructed the Holocene vegetation over Scotland using the REVEALS model for seven sites across the Scottish mainland, resulting in a maximal forest extent by 6.7 ka cal BP (Fyfe et al., 2013). The data presented here show an earlier woodland cover decline around 7.2 ka cal BP and a larger proportion of coniferous species in the forest composition. In comparison to the sites of Fyfe et al., the woodland cover shows to be relatively low, with an open heather

landscape prevailing during the period under study, which can be attributed to the relatively high elevation of the study area.

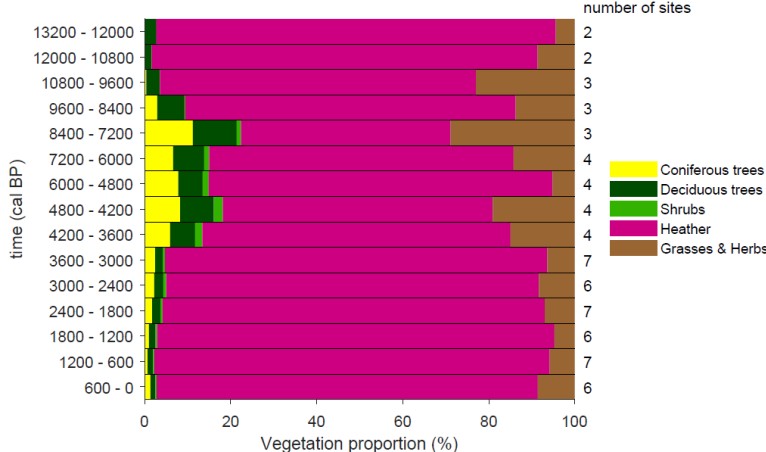

**Figure 6: Reconstructed vegetation proportions for the study area using the REVEALS model, based on seven pollen cores.**

### 3.2 Model calibration

Point-by-point calibration resulted in poor correspondence between the modelled and observed peat thickness. As a
consequence, the model parameters were calibrated based on the mean peat thickness per topographic class. In total, nine topographic classes were constructed by classifying all coring locations based on the slope and the hillslope curvature. The best fitting parameter combination results in an acrotelm thickness of 10 centimetres, an oxic decomposition rate at 10°C of 2.15 percent year$^{-1}$ and an anoxic decomposition rate at 10°C of 0.24 percent year$^{-1}$, which corresponds to an oxic/anoxic decomposition ratio of 9. These values correspond largely to those reported in the literature (Ballard et al., 2011; Clymo, 1984;
Wu, 2012; Yu et al., 2001). The RMSE on the mean peat thickness for the best fitting parameter combination is 9.53 centimetres (fig. 7).

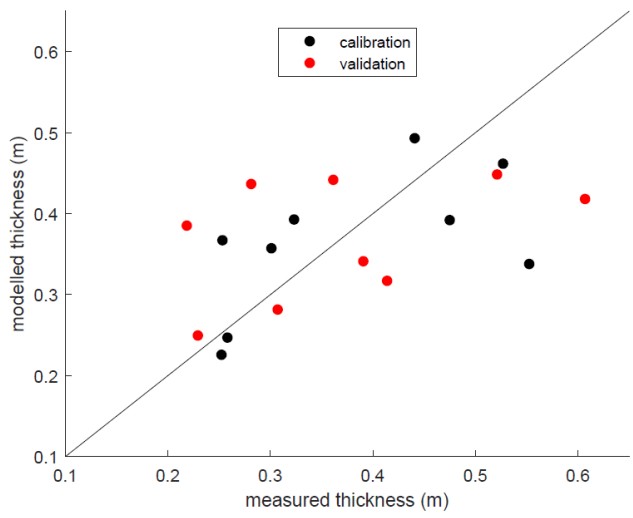

**Figure 7: Modelled and measured mean peat thickness per topographic class for both the calibration and validation transects.**

### 3.3 blanket peatland development

The calibrated model was run to simulate the long-term blanket peatland development since 12 ka BP for the 56 hillslope transects using the calibrated parameter values. The reconstructed land cover history (fig. 6) is used as vegetation evolution throughout the simulations. Overall, the model simulations indicate that mean peat accumulation rates were low until 9.5 ka BP, with small variations between the different gridpoints (fig. 8). Later, the accumulation rates increased and were high during two phases in the early Holocene, 9.5 – 8.5 ka BP and 8 – 6.5 ka BP. From 6 ka BP to 2 ka BP, the rates were relatively stable

and slightly positive on average. A long-term decrease in accumulation rates occurred between 2 ka BP and 1 ka BP, which increased again to positive values around 0.5 ka BP. The mean peat and carbon accumulation rate over all gridpoints and for the entire studied period is $0.03*10^{-3}$ m year$^{-1}$ and 1.79 g C m$^{-2}$ year$^{-1}$. The maximal mean peat and carbon accumulation rate over the entire studied period is $0.18*10^{-3}$ m year$^{-1}$ and 11.95 g C m$^{-2}$ year$^{-1}$ and occurs at 7050 years BP.

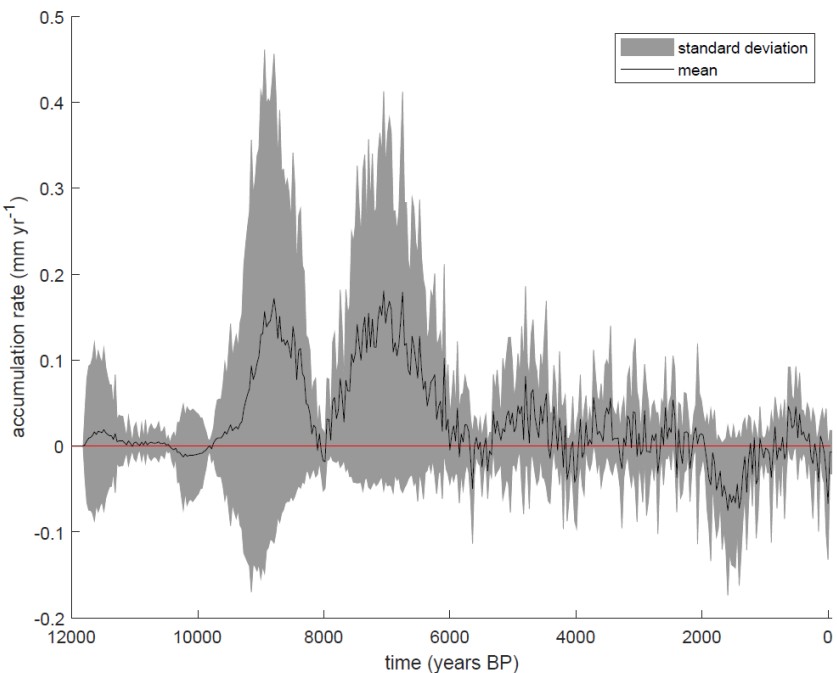


**Figure 8: Simulated mean peat accumulation rate and standard deviation for all studied hillslope transects (n = 56).**

Figure 9 indicates the evolution of the mean peat thickness and corresponding organic carbon mass over all hillslope transects. The mean peat thickness reaches a maximal value of 0.36 metres around 2 ka BP and declines slightly afterwards to a current value of 0.33 metres or 22.04 kg C m$^{-2}$. Overall, the peatland development occurs mostly before 6 ka BP and shows limited

variations afterwards, with a slight decline in mean peat thickness between 2 ka BP and 1 ka BP. When only the model gridpoints with a peat cover are considered, the maximal value for the modelled mean thickness is 0.66 metres, declining to

0.61 metres nowadays or 40.25 kg C m$^{-2}$. In total, the model simulates a peat cover at 61 percent of the coring locations, which is comparable to the coring data, which result in 57 percent peat cover.

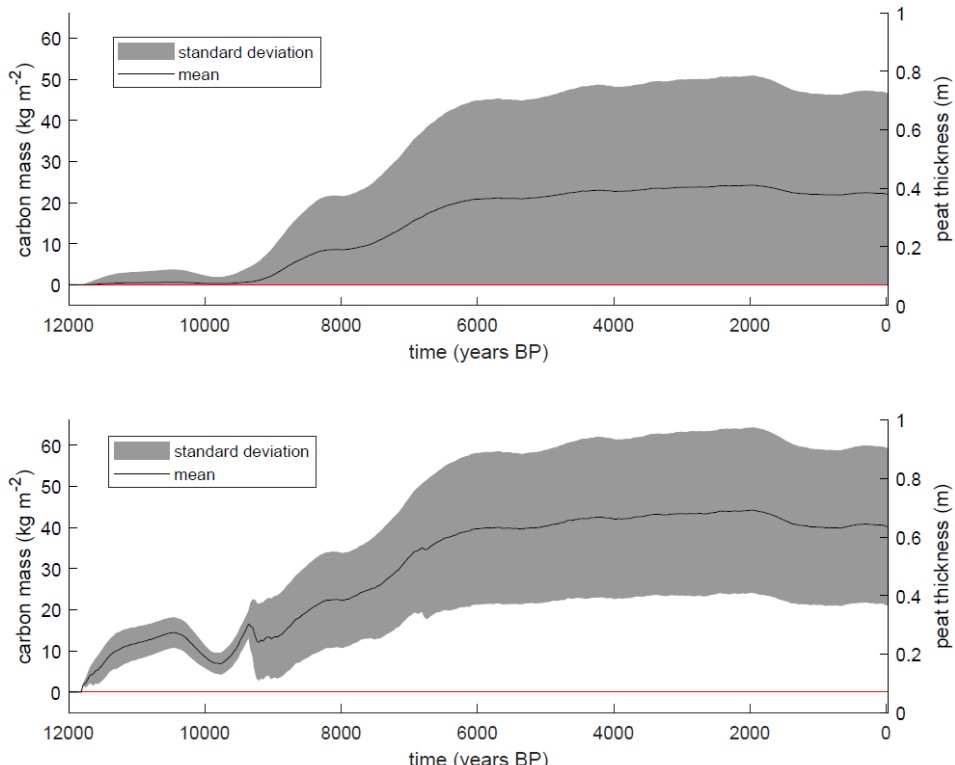

 **Figure 9: Simulated mean peat thickness/carbon mass and standard deviation for all gridpoints (top) and for the gridpoints with a peat cover (bottom).**

Although the model calibration is solely based on the current peat thickness, the timing of peatland development can be evaluated based on radiocarbon dating of peat profiles. Overall, the model simulations indicate that peat growth initiated at most locations between 9.75 ka BP and 6.75 ka BP. The basal radiocarbon dates (n = 47) show a more diffuse pattern for the Scottish upland areas (fig. 10a). When only considering those dates for which the radiocarbon sample was taken at the bottom of the peat column (n = 20), excluding the sites for which the initiation date was estimated by the extrapolation of an age-depth model to the bottom of the peat core, the probability density function shifts to older initiation ages and corresponds much better with the simulated dates (fig.10b).

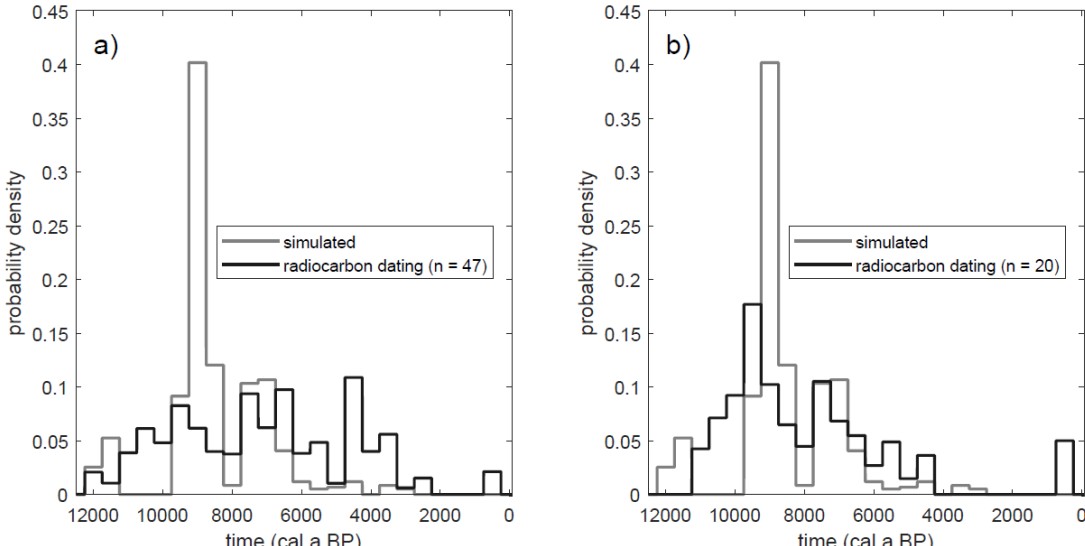

Figure 10: Probability density function of the peat growth initiation dates based on the model simulation and the radiocarbon dating database for Scottish upland areas (above 300 metres a.s.l., see appendix A2) using a bin width of 500 years. a) all dates. b) All dates, excluding those for which a date was obtained by extrapolating an age-depth model to the bottom of the peat column.

## 3.4 Sensitivity analysis

To study the model sensitivity to variations in parameter values a sensitivity analysis is carried out. In total, seven parameters are varied over the range as mentioned in the literature. The parameters broadly cluster in two groups, peat properties and environmental parameters (Table 1). Each parameter is varied stepwise, while all other parameters are kept at the standard value. The sensitivity is evaluated as the current mean peat thickness over all gridpoints after a simulation over a 12,000 year-period using the pollen-based climate and land cover variations as environmental boundary conditions (fig. 11). Overall, the model shows to be most sensitive to the peat decomposition rate, mean annual temperature and woodland cover. The peat thickness shows no sensitivity towards changes in catotelm conductivity, probably, because the low conductivity values of the catotelm anyhow result in slow drainage compared to other components of the hillslope hydrology and thus in quasi permanent water saturation of the catotelm. The acrotelm conductivity shows the same behaviour except for the lowest value. The acrotelm is under oxic conditions for most of the simulated values. Only for the lowest conductivity value, the water table rises above the catotelm-acrotelm boundary, resulting in lower decomposition rates and a higher mean peat thickness.

Table 1: Overview of the parameters used in the parameter sensitivity test, listing the standard value and the range over which the parameter is changed.

| Parameter | Standard value | Minimum value | Maximum value | References |
|---|---|---|---|---|
| **Peat properties** | | | | |

| | | | | |
|---|---|---|---|---|
| Acrotelm saturated hydraulic conductivity (m s$^{-1}$) | $1*10^{-3}$ | $1*10^{-5}$ | $1*10^{-2}$ | (Cunliffe et al., 2013; Holden et al., 2011; Ingram, 1983) |
| Catotelm saturated hydraulic conductivity (m s$^{-1}$) | $1*10^{-6}$ | $1*10^{-9}$ | $3*10^{-6}$ | (Cunliffe et al., 2013; Dai and Sparling, 1973; Holden and Burt, 2003; Ingram, 1983; Rosa and Larocque, 2008) |
| Acrotelm thickness (m) | 0.1 | 0.05 | 0.5 | (Ballard et al., 2011; Belyea and Malmer, 2004; Clymo, 1984) |
| Oxic decomposition rate at 10°C (% yr$^{-1}$) | 2.15 | 1 | 5 | (Kleinen et al., 2012; Lucchese et al., 2010; Malmer and Wallen, 2004; Wu, 2012; Yu et al., 2001) |
| **Environmental parameters** | | | | |
| woodland cover (%) | REVEALS-based land cover reconstruction (fig. 6) | 0 | 100 | (Hunter, 2016) |
| Mean annual temperature (°C) | 6.2 | -50% | +50% | Standard value based on the mean annual temperature for Braemar for the period 1890 – 1919 (earliest data available) (MetOffice, 2012) |
| Mean annual precipitation (mm yr$^{-1}$) | 900 | -75% | +75% | Standard value based on the mean annual precipitation for Braemar for the period 1890 – 1919 (earliest data available) (MetOffice, 2012) |

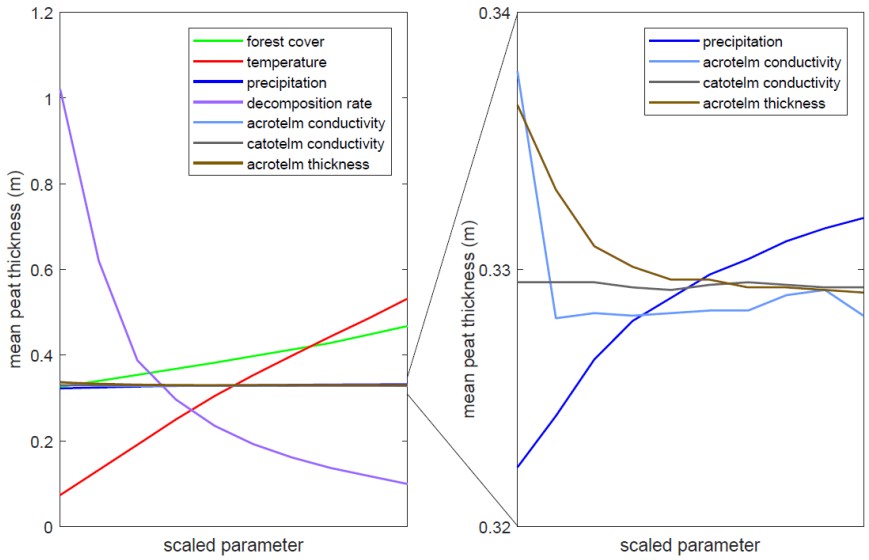

**Figure 11: Mean simulated peat thickness for all variables used in the parameter sensitivity test.**

The climate sensitivity of a model gridpoint appears to be dependent on the presence of a peat layer. Overall, the percentage of model gridpoints covered by peat shows to be more sensitive to precipitation changes than the mean peat thickness (fig. 12, fig. 13). This might be a result of the diplotelmic representation of the peat profile. The strong difference in saturated hydraulic

conductivity for the two peat horizons results in minimal water table changes when the precipitation amount is varied. In other words, the use of a diplotelmic model results in a water table which fluctuates only slightly around the acrotelm-catotelm boundary. As a result, the peat accumulation rates and resulting peat thickness are not very sensitive to precipitation changes. However, this does not mean that the water table is located at the same depth for all gridpoints. Depending on the local hillslope topography, some locations will be fully saturated for most of the time while at other locations, the water table will be located

much lower in the peat profile. For the substrate, only a single saturated hydraulic conductivity value is used, which results in a more sensitive response to precipitation changes for the gridpoints which do not have a peat cover. Overall, the highest peat thickness and percentage peat cover can be found for the scenarios with a high temperateture and precipitation amount.

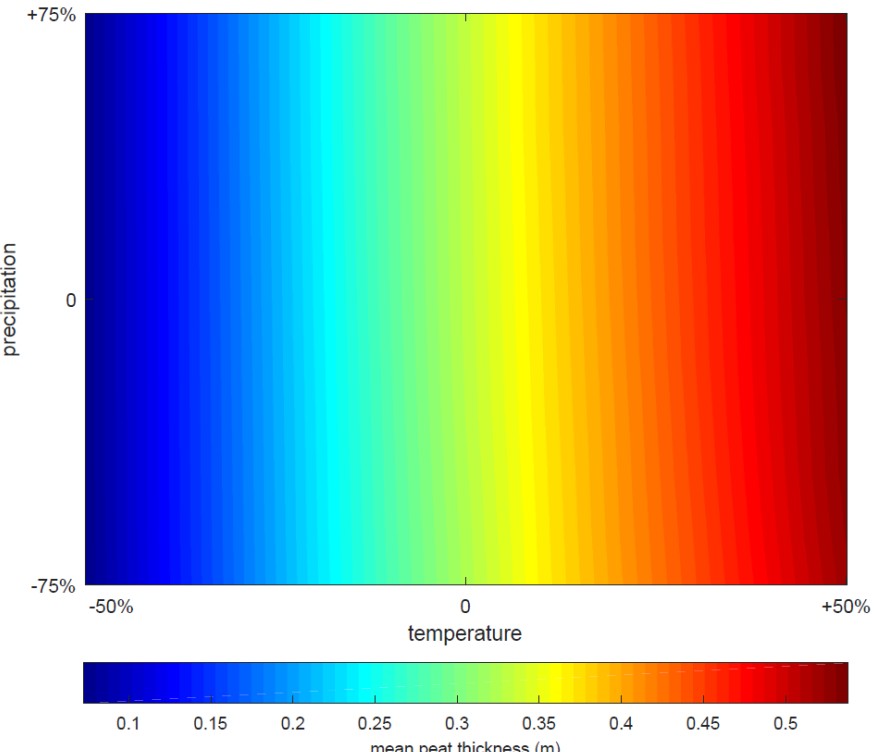

**Figure 12: Mean peat thickness for all combinations of temperature and precipitation changes.**

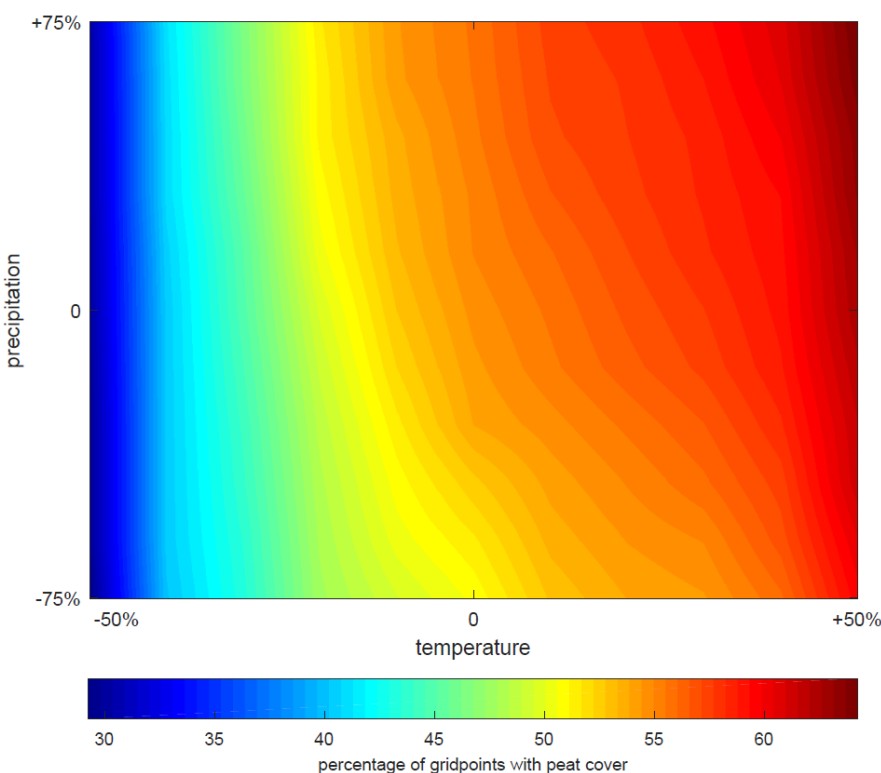

**Figure 13: Percentage of all model gridpoints with a peat cover of at least 10 centimetres for all combinations of precipitation and temperature variations.**

## 4 Discussion

Point-by-point calibration of the hillslope model resulted in a poor correspondence between modelled and observed peat thickness. Using the mean peat thickness per topographic class however allowed to calibrate the model with a RMSE below 10 centimetres (fig. 7). This indicates that a spatial peatland model with a simplified representation of the peat profile is unable to capture the local variability, but can replicate the general peatland evolution on the landscape scale. Similar model behaviour has been found in sediment erosion modelling, where point-by-point comparison yields poor correspondence but where the

mean value per topographic class performs sufficiently well (Peeters et al., 2006). The calibrated acrotelm thickness of 10 centimetres fits well within the range of 5-50 centimetres mentioned in the literature (Ballard et al., 2011; Belyea and Clymo, 2001; Clymo, 1984). The same holds true for the calibrated oxic decomposition rate at 10°C of 2.15 percent per year, where values between 0.25 and 7 percent per year can be found in the literature (Kleinen et al., 2012; Lucchese et al., 2010; Malmer and Wallen, 2004; Wu, 2012; Yu et al., 2001). In contrast, the calibrated anoxic decomposition rate at 10°C of 0.239 percent

per year is relatively high and exceeds values from other studies which find rates between $1.6*10^{-3}$ and $2.6*10^{-2}$ percent per year. The high calibrated decomposition rates can be attributed to the fact that these rates within the model do not only encompass peat decomposition within the soil profile, but also other processes which lead to decrease in peat thickness in the

field such as particulate organic carbon export through gully development and shallow mass movements. These processes are not represented in the model but affect the peat thickness as it is measured in the field, leading to higher decomposition rates

in the model calibration. The MILLENNIA peatland model contains a peat erosion module, calculating the total organic carbon (TOC) export based on the local water table depth and hillslope runoff depth (Heinemeyer et al., 2010). However, this approach is not able to discriminate between the different erosion processes observed in the field and would require a high amount of additional data. As studies on soil erosion have demonstrated that a model complexity reduction is necessary to reduce the model uncertainty when applied at the landscape scale, peat erosion is not included in the model (Jetten et al., 2003; Van

Rompaey and Govers, 2002). As a consequence, the calibrated decomposition rates must be regarded as a model parameter encompassing a range of processes removing peat mass from a certain location rather than simply the decomposition of organic matter within the peat profile itself.

Overall, the peatland model is not able to simulate the high peat thickness values (larger than 1.5 metres), observed at some locations in the landscape. This can be partially attributed to the relatively high calibrated decomposition rates (fig. 14). A

second reason might the fact that local depressions within the hillslope topography were filtered out to enable the use of the Boussinesq-equation for the simulation of the subsurface flow. Since local depressions showed to contain thick peat deposits in the field, the filtering procedure on the hillslope topography reduces the potential of modelling high peat thickness values at these locations.

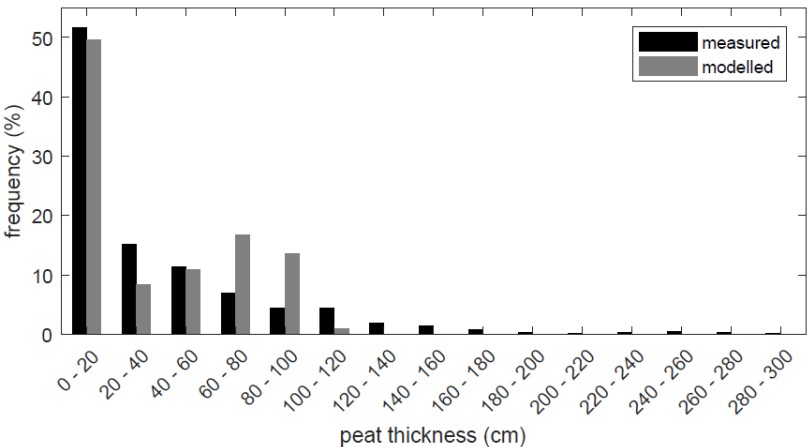

**Figure 14: Frequency distribution of the measured and modelled peat thickness for all studied hillslope transects (n = 56).**

The range of simulated peat accumulation rates shows to be realistic, with periods of high mean accumulation rates coinciding with periods of temperature increase (fig. 3, fig. 8). More specific, the mean peat accumulation rates were high during the periods 10 ka – 8.5 ka BP where the mean annual temperature increased with 3.74°C, resulting in a mean peat accumulation rate over all modelled locations for the entire period of $0.064*10^{-3}$ m yr$^{-1}$ or 4.24 g C m$^{-2}$ yr$^{-1}$ and 8 ka – 6.5 ka BP, with a

temperature increase of 2.34°C and a mean peat accumulation rate of $0.104*10^{-3}$ m yr$^{-1}$ or 6.88 g C m$^{-2}$ yr$^{-1}$. It appears to be the temperature increase, rather than the temperature itself, which drives peat growth. The increased biomass production due

to the temperature increase outweighs the lowering in water table height caused by higher evapotranspiration rates and creates an imbalance between production and decomposition, leading to positive accumulation rates and a peat thickness increase.

In contrast to existing cohort models, which have shown to be capable of capturing local variations in dynamics within the peat profile, the relatively simple diplotelmic model presented here cannot reproduce the local dynamics with the same degree of detail (Frolking et al., 2010; Heinemeyer et al., 2010; Morris et al., 2012). However, the simple representation of the model domain leads to a decrease in computation time which allows the application of the model over large spatial and temporal domains. In combination with the pollen-based climate and land cover reconstructions, it allows to study peatland development on the landscape scale, rather than at the scale of a single peat bog or peat profile as is often the case for the cohort models, allowing to answer different research questions.

## 4.1 Peat growth initiation

Based on the model simulations, the peat growth initiation dates cluster mostly within the period 9.75 ka – 6.25 ka BP (fig. 10), which corresponds largely to the Atlantic period, which is mentioned by other studies based on field data (Ellis and Tallis, 2000; Tipping, 2008). The database of basal radiocarbon dates shows a wider spread of initiation dates for upland Scotland between 12.25 ka BP and 3.25 ka BP. However, a majority of the available basal dates within the dataset is based on a radiocarbon dating higher in the peat profile, which is extrapolated to the bottom of the core using an age-depth model. One could question whether this extrapolation is justified. The deeper parts of the peat profile are older and as a result, a smaller fraction of the originally deposited biomass will remain. This leads to lower reconstructed accumulation rates further back in time. The reconstructed rates based on the available radiocarbon dates for the study area show a clear decreasing trend when going further back in time (fig. 15, see appendix A2 for dating details). When extrapolating to the bottom of a peat profile in order to obtain an initiation age, the accumulation rate of the above-lying part of the peat profile is used to extrapolate over the lowest part of the profile, which is older and has a lower fraction of remaining peat mass. As a result, these extrapolated ages will be biased towards younger basal ages. This effect increases with increasing distance between the bottom of the peat profile and the deepest radiocarbon date. When excluding the extrapolated basal ages from the analysis, the probability density function shifts towards older ages, clustering between 11.25 ka BP and 4.25 ka BP and corresponding much better with the probability density function of the model simulations (fig. 10). Here, both the radiocarbon dates as well as the model simulations show peaks in peat growth initiation in the periods 10.25 – 8.25 ka BP and 7.75 – 6.25 ka BP, which coincide with the periods of temperature increase. Overall, the simulations show broadly the same pattern as the radiocarbon dates, but with a more pronounced difference between periods. The spikes in the simulated initiation dates might be attributed to the use of a single set of parameter values for the entire study area resulting in a relatively similar response of many model gridpoints to the changing environmental conditions during the early-Holocene. The more diffuse probability density function for the radiocarbon dates might thus, at least partially, be ascribed to local heterogeneity. Additionally, as the available basal radiocarbon dates come from areas all over Scotland, the probability density curve is likely to include regional differences in peatland initiation ages as well.

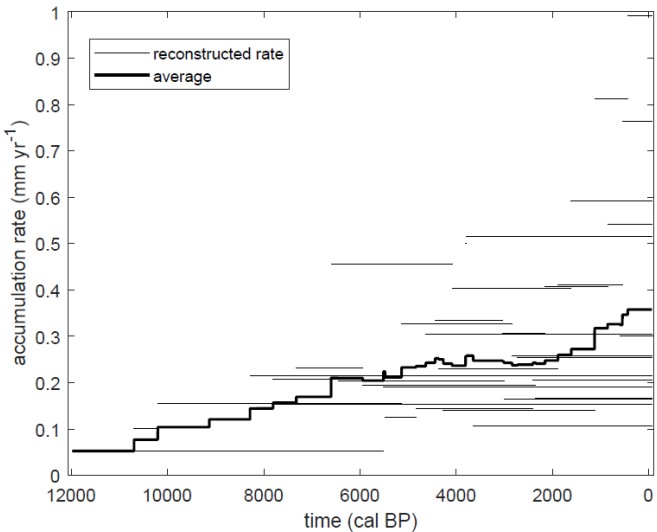


**Figure 15: Reconstructed peat accumulation rates based on all available radiocarbon dates within the study area and mean peat accumulation rate .**

A study for the British Isles based on an envelope climate model for blanket peatlands finds a contraction in the area suitable for blanket peatland development in eastern Scotland since 6 ka BP and other studies find post-6 ka accumulation rates to be
relatively low (Gallego-Sala et al., 2016; Simmons and Innes, 1988). In this study, accumulation rates decrease from 8 ka to 6 ka BP onwards (fig. 7). Overall, the mean accumulation rate remains positive until approximately 2 ka BP, but never reaches the high values which occurred during the early Holocene. This results in a slowdown in the peatland development and carbon storage after 6 ka BP (fig. 9). The asymptotic behaviour of the peat thickness evolution at the landscape scale, stabilising after 6 ka BP, is also found in other studies modelling long-term peatland development at the local scale. While the modelled peat
thickness trajectory is dependent on the specific conditions (climate, land cover, topography, …), it is clear that the carbon-sequestering potential of a peatland has it limits at millennial timescales, as the balance between biomass production and decomposition comes in equilibrium with the environmental conditions (Frolking et al., 2010; Heinemeyer et al., 2010).

The conclusion that the blanket peatland development in the Upper Dee area can be attributed to a climate warming, independent of an increase in precipitation, as demonstrated by the sensitivity analysis, is in line with a study by a Morris et
al., who compared a large dataset of peatland initiation dates across the globe with GCM paleoclimate simulations, concluding that peatland initiation in formerly glaciated areas can be attributed to rising growing season temperatures (Morris et al., 2018). Additionally, a recent study on buried peat layers indicates that in northern latitudes (>40°N) peat growth is extensive during warm periods such as the last interglacial and the MIS 3 interstadial (57 - 29 ka) (Treat et al., 2019). It is clear that anaerobic conditions are required for the development of peat soils. However, regional climatic changes towards wetter conditions do
not seem to be necessary for blanket peatland initiation. Apparently, local factors driving the hydrology such as hillslope topography, soil properties, etc. will determine where anoxic conditions will establish to enable blanket peatlands to develop (Morris et al., 2018).

The model simulations do not support the original hypothesis on the origin of the blanket peatlands, linking the peatland development to a deforestation-driven change in hillslope hydrology (Moore, 1973). Firstly, both the available basal radiocarbon dates and the simulated initiation dates indicate a shift towards peat soils during a period of increasing or stable woodland cover (fig. 6, fig. 10). Secondly, the parameter sensitivity analysis indicates that a decrease in tree cover, either by natural or anthropogenic causes, decreases the peat growth potential because the decrease in evapotranspiration due to a loss of tree cover is outweighed by the reduction in biomass production under the environmental conditions present in the study area. Tipping (2008) studied the Holocene blanket peatland development in five upland and northern sites in Scotland using a combination of geomorphic, archaeological and radiocarbon data, resulting in the hypothesis that blanket peatlands were common over large parts of the Scottish Highlands within the first few millennia of the Holocene either due to rapid soil development or by climatic changes (Tipping, 2008). This study supports the hypothesis of Tipping, as shown by the model simulations and peatland initiation dates and provides evidence that a changing climate (increasing mean annual temperature) was the main driver of blanket peatland development.

Although the simulated mean peat accumulation rates remain at low levels after 6 ka BP, this does not mean peat profiles are unable to develop but rather that the peatlands at landscape scale are in dynamic equilibrium with the stabilising Holocene climate. This can be demonstrated using a forced model simulation where all peat soils are removed at 6 ka BP. The resultant simulated peat thickness evolution indicates that peat starts to develop immediately after the peat removal (fig. 16). After approximately 1500 years, the mean peat thickness over the study area reaches again the values of the standard model run. This indicates that the model can simulate peatland regeneration at locations which are impacted by a removal of the peat cover either by natural processes (e.g. by shallow mass movements, gullies) or following anthropogenic peat cutting.

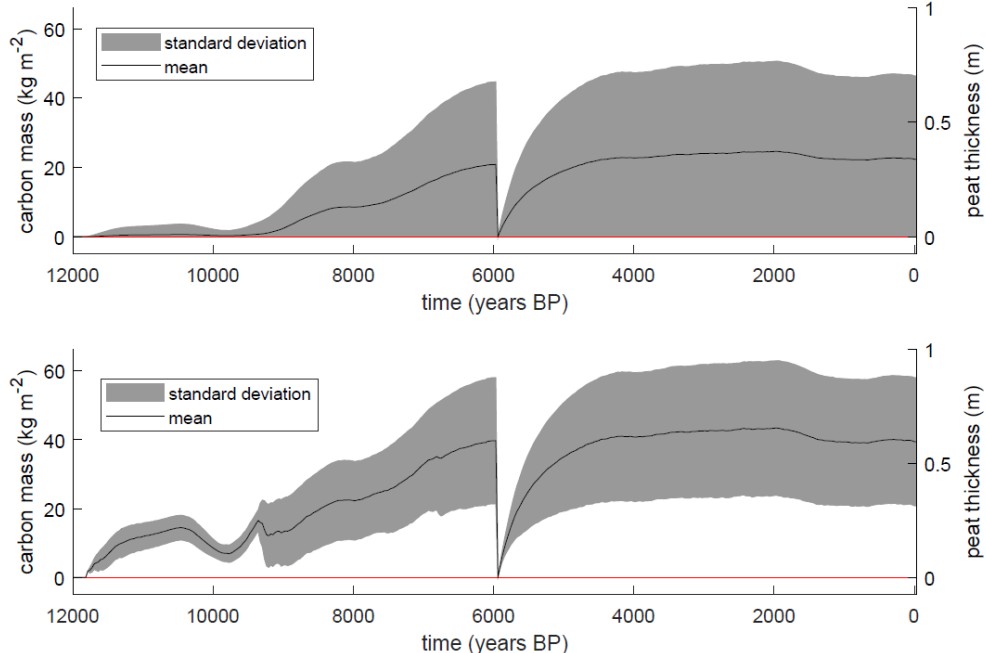

**Figure 16: Simulated mean peat thickness/carbon mass and standard deviation for all gridpoints (top) and for the gridpoints with a peat cover (bottom), with a removal of all peat cover within the study area at 6 ka BP.**

## 5 Conclusion

A new process-based model was presented to study long-term blanket peatland development along hillslopes. The simulations for the past 12,000 years indicate that a relatively simple diplotelmic model is able to capture long-term peatland dynamics on the landscape scale. However, point-by-point comparison still holds poor results, which can be attributed to the use of a single set of calibrated parameters and the idealised representation of the model domain. Overall, both the field data and model simulations indicate that the blanket peatlands in the Upper Dee area developed mostly during the Atlantic period, with a peak in peat growth initiation dates around 9 ka BP. The timing of peatland initiation together with the results of the sensitivity analysis support the hypothesis of a climate-driven origin of the blanket peatlands in the Scottish highlands, where the peatland development shows to be driven by a long-term regional warming trend during the early-Holocene. A higher woodland cover leads to an increase in peat growth potential, contradicting the original hypothesis of Moore (1973), which identified deforestation as a potential driver of blanket peatland development. In more recent periods, the relatively stable climate and land cover within the study area since 6 ka BP result in a stabilisation of the peatland development, indicating that the study area served as a terrestrial carbon sink mainly during the Atlantic period and has stabilised during the late-Holocene.

**Appendices**

**A1: MIDAS (Met office Integrated Data Archive System) weather stations used for the construction of the regression equations for orographic temperature and precipitation corrections.**

| Station name | MIDAS station code | Elevation (metres a.s.l.) | Latitude (degrees North) | Longitude (degrees West) |
|---|---|---|---|---|
| Braemar (irrigation farm) | 14938 | 323 | 57.012 | 2.745 |
| Dalwhinnie | 14769 | 361 | 56.928 | 3.403 |
| Corgarff | 144 | 400 | 57.165 | 2.733 |
| Forest Lodge No. 2 | 15190 | 305 | 56.847 | 3.001 |
| Trinafour | 15183 | 268 | 56.751 | 2.856 |
| Glenshee Lodge | 225 | 335 | 56.799 | 3.111 |
| Pitcarmick | 15251 | 198 | 56.694 | 2.627 |
| Inschriach | 14783 | 213 | 57.144 | 1.954 |

**A2: Radiocarbon dating results. Calibrated ages were calculated using the Oxcal 4.3 software and the IntCal13 calibration curve (Bronk Ramsey, 2009; Reimer et al., 2013).**

| Sample ID | Lab-code | Conventional age (BP) | Calibrated age (cal a BP ± 1σ) | Longitude (degrees) | Latitude (degrees) | Dated material | Sample depth (m) | Total peat depth (m) | Reference |
|---|---|---|---|---|---|---|---|---|---|
| DEEH1P3 | RICH-26352 | 594 ± 23 BP | 599 ± 26 cal a BP | -3.5117 | 56.9711 | bulk peat | 0.20 | 0.20 | This study |
| Allt Connie H4P5-1 | RICH-25414 | 941 ± 27 BP | 855 ± 39 BP | -3.5667 | 56.9468 | Plant remain | 0.5 | 1.77 | This study |
| Allt Connie H4P5-2 | RICH-25429 | 2145 ± 29 BP | 2155 ± 79 BP | -3.5667 | 56.9468 | Bulk peat | 1.03 | 1.77 | This study |
| Allt Connie H4P5-3 | RICH-25435 | 2899 ± 30 BP | 3036 ± 52 BP | -3.5667 | 56.9468 | Bulk peat | 1.3 | 1.77 | This study |
| Allt Connie H4P5-4 | RICH-25415 | 3827 ± 31 BP | 4440 ± 69 cal a BP | -3.5667 | 56.9468 | bulk peat | 1.70 | 1.77 | This study |
| DEEH8P7 | RICH-26330 | 2430 ± 25 BP | 2724 ± 87 cal a BP | -3.6235 | 57.0005 | bulk peat | 0.65 | 0.71 | This study |
| DEEH8P12-1 | RICH-26327 | 2383 ± 25 BP | 2408 ± 55 BP | -3.6258 | 56.9979 | Bulk peat | 0.51 | 0.94 | This study |
| DEEH8P12-2 | RICH-26349 | 4265 ± 27 BP | 5480 ± 42 cal a BP | -3.6258 | 56.9979 | bulk peat | 0.86 | 0.94 | This study |
| DEEH8P13-1 | RICH-26350 | 4764 ± 27 BP | 5514 ± 52 BP | -3.6265 | 56.9973 | Bulk peat | 1.06 | 1.40 | This study |
| DEEH8P13-2 | RICH-26329 | 7543 ± 31 BP | 11979 ± 94 cal a BP | -3.6265 | 56.9973 | bulk peat | 1.21 | 1.40 | This study |
| DEEH8P16 | RICH-26334 | 6994 ± 31 BP | 9132 ± 55 cal a BP | -3.6278 | 56.9958 | wood | 1.22 | 1.42 | This study |
| DEEH9P7 | RICH-26351 | 3241 ± 25 BP | 3649 ± 40 cal a BP | -3.4847 | 57.0352 | bulk peat | 0.38 | 0.4 | This study |
| DEEH10P8-1 | RICH-26331 | 2333 ± 25 BP | 2347 ± 20 BP | -3.4682 | 57.0499 | Bulk peat | 0.4 | 1.42 | This study |
| DEEH10P8-2 | RICH-26333 | 5184 ± 28 BP | 7324 ± 51 cal a BP | -3.4682 | 57.0499 | bulk peat | 1.10 | 1.42 | This study |
| DEEH13P7 | RICH-26324 | 7350 ± 30 BP | 8287 ± 60 cal a BP | -3.5606 | 56.9251 | bulk peat | 1.77 | 1.80 | This study |
| DEEH15P1 | RICH-26326 | 3865 ± 24 BP | 4633 ± 60 cal a BP | -3.5305 | 56.9106 | wood | 1.33 | 1.43 | This study |
| DEEH23P13-1 | RICH-26328 | 2756 ± 25 BP | 2842 ± 36 BP | -3.5656 | 57.0686 | Bulk peat | 0.75 | 2.34 | This study |
| DEEH23P13-2 | RICH-26332 | 4461 ± 27 BP | 5137 ± 97 BP | -3.5656 | 57.0686 | Bulk peat | 1.5 | 2.34 | This study |
| DEEH23P13-3 | RICH-26335 | 9029 ± 39 BP | 10697 ± 33 cal a BP | -3.5656 | 57.0686 | bulk peat | 2.29 | 2.34 | This study |
| DEEH42P2 | RICH-26325 | 3519 ± 28 BP | 3806 ± 42 cal a BP | -3.3981 | 56.9023 | bulk peat | 1.99 | 2.00 | This study |
| Allt Connie AC200-1 | RICH-25434 | 528 ± 28 BP | 549 ± 33 BP | -3.5648 | 56.9451 | Bulk peat | 0.47 | 1.59 | This study |
| Allt Connie AC200-2 | RICH-25430 | 1939 ± 29 BP | 1888 ± 35 BP | -3.5648 | 56.9451 | Bulk peat | 1.02 | 1.59 | This study |
| Allt Connie AC200-3 | RICH-25433 | 3882 ± 31 BP | 4455 ± 114 cal a BP | -3.5648 | 56.9451 | wood | 1.58 | 1.59 | This study |
| Bynack Burn pollen core-1 | RICH-22684 | 1570 ± 31 BP | 1466 ± 40 BP | -3.6689 | 56.9497 | Bulk peat | 1.08 | 2.77 | (Hunter, 2016) |
| Bynack Burn pollen core-2 | RICH-22690 | 2008 ± 28 BP | 1956 ± 35 BP | -3.6689 | 56.9497 | Bulk peat | 2.16 | 2.77 | (Hunter, 2016) |
| Bynack Burn pollen core-3 | RICH-22687 | 3006 ± 34 BP | 3409 ± 76 cal a BP | -3.6689 | 56.9497 | bulk peat | 2.68 | 2.77 | (Hunter, 2016) |
| Geldie Burn pollen core-1 | RICH-22689 | 1688 ± 28 BP | 1598 ± 40 BP | -3.6667 | 56.9789 | Bulk peat | 1.00 | 3.14 | (Hunter, 2016) |
| Geldie Burn pollen core-2 | RICH-22685 | 3729 ± 33 BP | 4073 ± 59 BP | -3.6667 | 56.9789 | Bulk peat | 2.00 | 3.14 | (Hunter, 2016) |
| Geldie Burn pollen core-3 | RICH-22686 | 5510 ± 38 BP | 6580 ± 48 cal a BP | -3.6667 | 56.9789 | bulk peat | 3.02 | 3.14 | (Hunter, 2016) |
| Geldie Lodge pollen core-1 | GU-17252 | 2880 ± 35 BP | 3010 ± 58 BP | -3.7181 | 56.9625 | Bulk peat | 0.51 | 1.48 | (Paterson, 2011) |
| Geldie Lodge pollen core-2 | GU-17254 | 5540 ± 35 BP | 7745 ± 64 cal a BP | -3.7181 | 56.9625 | bulk peat | 1.43 | 1.48 | (Paterson, 2011) |
| Luibeg H200-1 | RICH-25412 | 386 ± 26 BP | 437 ± 59 BP | -3.6457 | 57.0444 | Plant remain | 0.5 | 1.50 | This study |
| Luibeg H200-2 | RICH-25427 | 1200 ± 28 BP | 1127 ± 45 BP | -3.6457 | 57.0444 | Plant remain | 1.06 | 1.50 | This study |
| Luibeg H200-3 | RICH-25436 | 3603 ± 32 BP | 4411 ± 113 cal a BP | -3.6456 | 57.0444 | bulk peat | 1.45 | 1.50 | This study |


**Data availability**

The supplementary data to this article consist of two datasets. A list of all soil corings, including the coring location, elevation
and measured peat depth is available online at: http://dx.doi.org/10.17632/pxszz2wzny.1. The detailed stratigraphic
descriptions of each soil coring is available online at: http://dx.doi.org/10.17632/ms484mrjj5.1.

**Author contribution**

The conceptualisation and methodology development of this project was carried out by WS, NB and GV. The field work was
performed by WS, NB and GV. WS carried out the lab work, developed the model code and performed the model simulations.
GV and NB supervised the research. The writing of the manuscript was carried out by WS, NB and GV.

**Competing interests**

The authors declare that they have no conflict of interest.

**Acknowledgements**

Ward Swinnen holds a PhD grant of the FWO – Research Foundation Flanders (application 1167019N). This research is part
of a project funded by the FWO (application G0A6317N). The authors thank Mar estate, Mar lodge estate and Invercauld
estate for the permission to access the area. The authors thank Teun Daniëls, Sofie De Geeter, Yasmine Hunter, Ellen Jennen,
Vincent Lenaerts and Remi Swinnen for their assistance during the field campaigns. Danny Paterson and Richard Tipping are
thanked for sharing their pollen data from the Upper Dee catchment. Pollen data of Birks (1969) and Huntley (1994) were
extracted from the European Pollen Database (EPD; http://www.europeanpollendatabase.net/) and the work of the data
contributors and the EPD community is gratefully acknowledged.

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
