# Peer review of "Modelling long-term blanket peatland development in eastern Scotland."

_Biogeosciences, 2019_

## Referee Comment (RC1) · Anonymous Referee #1 · 5 Jul 2019

Although the manuscript could be published in its present form, I recommend a minor revision to make reported results reproducible and to clarify the concept of the work.

Below are my reflections that authors might find useful as a source of ideas for improving the manuscript.

The blanket peatland architecture reconstructed along 56 hillslope transects is a valuable source of information for validating the models of peat accumulation. Are this data available from authors? Or is there any plan to make these data open for re-use? I recommend to add a brief section about data availability if authors are planning to make data open or available under some conditions.

Which numerical method did authors use for solving the Eq (1)? I recommend to add

a phrase directing a reader to the article where the numerical method used for solving the Eq (1) is described.

It seems to me that the changes in hillslope topography resulted from peat accumulation do not affect the water storage (S (x)) given by Eq (1), because this equation takes into account only the bedrock slope (i(x)) which is not affected by peat accumulation. If my understanding is correct, then S(x) is the maximum water storage that could be achieved under given climatic conditions and the bedrock slope (i(x)).

The model used by authors is based on the concept of impeded drainage [1-3] suggesting that geomorphological conditions (i.e. bedrock slope) determine the maximum peat depth under given climatic conditions. Therefore, it would be interesting to see if there is a significant correlation between the measured peat depth (averaged over the transect) and the bedrock slope (averaged over the transect). The lack of significant correlation may suggest that the observed range of variations in the bedrock slope does not lead to a dramatic difference in the S (averaged over the transect).

[1] Ingram, H.A.P.: Size and shape in raised mire ecosystems: a geophysical model. Nature 297, 300–303, 1982.

[2] Clymo, R. S.: The Limits to Peat Bog Growth, Philos. Trans. R. Soc. B Biol. Sci., 303(1117), 605–654, doi:10.1098/rstb.1984.0002, 1984.

[3] Alexandrov, G. A., Brovkin, V. A. and Kleinen, T.: The influence of climate on peatland extent in Western Siberia since the Last Glacial Maximum, Sci. Rep., 6, doi:10.1038/srep24784, 2016.

---

## Referee Comment (RC2) · Andreas Heinemeyer (Referee) · 26 Jul 2019

Overall quality of the discussion paper ("general comments")

This study is an interesting one. It tackles an important issue, our ability to model and predict how blanket bog peatlands evolved and (based on their current C balance) could behave under future climate change. It also highlights the limitation of peatlands to sequester carbon in the long-term: as the C balance of input vs decomposition decreases over time, the ability for net sequestration is limited. This issue is often overlooked and needs to be highlighted.

The authors could have added a few more modelling studies which already highlighted this issue (i.e. Heinemeyer et al., 2010; Frolking et al., 2010) – plenty of peatland development models show an asymptotic C accumulation over time, leveling off during the past few thousand years. However, this depends on the peatland conditions (climate and topography). A bit more context in the introduction and discussion would be beneficial.

Although the model is interesting there are a few points I raise (also see below). One important, related aspect is the recreated past climate. I do not think that the anomalies for temperature or rainfall are capturing past changes. Just think about the little ice age/medieval warm period etc. Maybe consider Heinemeyer et al„ 2010 and Morris et al., 2015 for some UK specific reconstructions. 8 mm difference in monthly rainfall seem somewhat meaningless. Also, the initial warming impact and wetting at the onset of the warming about 12k ya should be more pronounced. I suggest you consider some specific literature (although I acknowledge that actually data on this is not that easy to come by). I think the issue is mainly in using a large scale model output – actually differences are lost (same as when averaging current climate over large grid scales). I suggest you discuss this limitation in the light of the above concerns and publications.

Moreover, the lack of a link between runoff and erosion within a hill slope context seems very odd to me. I suggest you consider this but maybe I have missed something in the methods. I just cannot find a link – which is crucial to allow slope C accumulation (which is not only decomposition [water table x temperature] driven). But your calibration will lead to overcompensation because of an important C flux process missing in your overall C budget.

This previous point also relates to the possible issue that underlying bedrock slope might not relate directly to peat surface slope (depressions) – another reviewer already made a comment on this. Would be nice to see a comparison in this respect.

Very interesting to see such a high level of heather domination over time. Also the hill slope data are if general interest. Are those data to be made available? I suggest a section and/or doi for the data.

Scientific questions/issues ("specific comments")

I think my main concerns are some assumptions like: L129 The underlying bedrock is impermeable (drainage). In most cases it is, and that will likely affect water tables during dry periods. They do have hydraulic conductivity and porosity...but in line 147 you mention that the mineral layer has been assigned a hydraulic conductivity. Explain and be consistent throughout the manuscript.

L143 The hydrology model just assumes that the water table depth is 'always' near the surface. This is not true. I bet in 2018 the water tables even in deeper and normally wet Scottish blanket bog went down to 30 cm or even more.

L196 Litter input and quality (NPP) are affected by both, temperature and precipitation (Leith's equation). It is a bit odd to use the Moor House equation, which is purely temperature based. Maybe provide a critical assessment of doing so. Particularly the role of trees in peat depend on the water table depth (trees evaporate a lot and can cause shrinkage and decomposition of peat – but the understanding about this is still somewhat limited). Maybe consider recent publications on Scottish afforested bog by Sloan et al (with the late Richard Payne as his supervisor).

L148 Snow evaporation during frosty and dry periods can be important (see Carroll et al., 2015 model description).

L168 AET is also dependent on rooting depth in relation to the water table depth (see Carroll et al., 2015 model description).

L207 It seems a bit odd to have a 2 phase Q10 which will cause a jumping up and down in steps (why not have a continuous change in Q10 – simple function?). Also, the Q10 question is very questionable indeed – apparent vs intrinsic & short-term vs long-term & with roots vs without roots - but that is a complex issue. It would be good to also provide references as to those two Q10 values.

Abstract:

[Figure]

Please add a bit more information on what kind of model you developed/tested (empirical/process) "spatially-explicit hill slope model".

Please add more information to what this means "peatland architecture was reconstructed". Currently this is unclear (did you take core samples/use a pollen and/or peat depth database?). Also, is 'assessed' a better word?

Please also add more information on what kind of climate data you used "climate reconstructions" e.g. mean annual precipitation/temperature . . . . The above points will help the reader grasp the complexity of what you did and put it into context for specific interests.

Introduction:

There are a lot of Gallego-Sala references when talking about basics of blanket bog. Is this the first time those explanations are made? I would have thought that Lindsay would have done this earlier and others before him. It is always good to go back to the original people.

I think it would be nice to mention the MILLENNIA (Heinemeyer et al., 2010) peatland model – coming out at the same time as HPM but for a UK context (so I suggest it is important in this context). This is also relevant when considering hill slope (see topography effects on temperature and hydrology, especially runoff and erosion - as in your study also using Garnett peat depth data for a validation at Moor House) and root C inputs. However, a valid point of criticism would be that it is a 2D model – so you are trying to do something better, which is great. However, you might also want to discuss what the disadvantages are of your simpler 2 depth model (root C input vs water table depth and porosity over depth affecting oxygen availability and thus decomposition – see Carroll et al. 2015 using the MILLENNIA model).

Finally, the empirical C input relationship with only temperature seems questionable when NPP is clearly dependent on both, temperature and precipitation – potential

**BGD**

evapotranspiration functions are commonly used as in the MILLENNIA model, Durham Carbon Model by Worrall et al. etc.).

Typos etc. ("technical corrections")

Field data: L99 Maybe better to use 'assessed' instead of reconstructed. L100 Soil cores were 'taken' along... L104 give details of the sections (what depth ranges were assessed – BD changes with depth).

Model Outline L131 Please add more information regarding "Boussinesqequation for the simulation of the hillslope hydrology".

Peat Initiation I like that there is a threshold for 'peat' based on C accumulation in the top mineral layer! Nice work. However, I suggest that a bit more information is needed again for the definition of the threshold (%Corg?).

Model Calibration I think it would be nice to see how all the parameters changed from default to calibrated. Possibly also how they compare to other publications (if applicable; e.g. HPD).

Results I think reporting dry bulk density values is best in g/cm3 – most models and field measurements in peatlands will show those units (unless I am mistaken).

---

## Author Comment (AC1) · 23 Aug 2019

**Author response to the interactive comment RC1 by an anonymous referee.**

In the text below, the authors respond to the comments given by referee 1. The comments of the referee are given as plain text, while the authors response is given in *italic*.

**Comments**

Although the manuscript could be published in its present form, I recommend a minor revision to make reported results reproducible and to clarify the concept of the work. Below are my reflections that authors might find useful as a source of ideas for improving the manuscript. Are this data available from authors? Or is there any plan to make these data open for re-use? I recommend to add a brief section about data availability if authors are planning to make data open or available under some conditions.

*It is the intention of the authors to make the soil coring dataset (i.e. coordinates of soil corings and stratigraphy) available by adding the data to the article as supplementary material. A short section on supplementary data will be added to the manuscript.*

Which numerical method did authors use for solving the Eq (1)? I recommend to add a phrase directing a reader to the article where the numerical method used for solving the Eq (1) is described.

*The numeric scheme used to discretize the Boussinesq equation consists of a first-order upwind finite difference scheme for the advection component and a forward in time – centred in space finite difference scheme for the diffusion component. This numerical method is, for instance, well described in Campforts & Govers 2015 (Campforts, B. & Govers, G., Keeping the edge: A numerical method that avoids knickpoint smearing when solving the stream power law., Journal of Geophysical Research: Earth Surface, 2015). We will update the manuscript to add this reference.*

It seems to me that the changes in hillslope topography resulted from peat accumulation do not affect the water storage (S (x)) given by Eq (1), because this equation takes into account only the bedrock slope (i(x)) which is not affected by peat accumulation. If my understanding is correct, then S(x) is the maximum water storage that could be achieved under given climatic conditions and the bedrock slope (i(x)).

The model used by authors is based on the concept of impeded drainage [1-3] suggesting that geomorphological conditions (i.e. bedrock slope) determine the maximum peat depth under given climatic conditions. Therefore, it would be interesting to see if there is a significant correlation between

the measured peat depth (averaged over the transect) and the bedrock slope (averaged over the transect). The lack of significant correlation may suggest that the observed range of variations in the bedrock slope does not lead to a dramatic difference in the S (averaged over the transect).

*There is indeed a relationship between the bedrock slope and the peat thickness measurements both for the individual coring locations as for the mean values per transect (see figures below). There is a clear decrease in the maximum observed peat thickness with increasing bedrock slope, or, thick peat layers cannot be found on steeper slopes. However, the large scatter indicates that thin peat layers or even the absence of a peat cover can be found for all slope values.*

[Figure]

*Figure 1: Scatterplot of the mean peat thickness per hillslope transect as a function of the mean bedrock slope per hillslope transect.*

[Figure]

*Figure 2: Scatterplot of the measured peat thickness per coring location as a function of the bedrock slope.*

**Suggested references by the referee**

[1] Ingram, H.A.P.: Size and shape in raised mire ecosystems: a geophysical model. Nature 297, 300–303, 1982.

[2] Clymo, R. S.: The Limits to Peat Bog Growth, Philos. Trans. R. Soc. B Biol. Sci., 303(1117), 605–654, doi:10.1098/rstb.1984.0002, 1984.

[3] Alexandrov, G. A., Brovkin, V. A. and Kleinen, T.: The influence of climate on peatland extent in Western Siberia since the Last Glacial Maximum, Sci. Rep., 6, doi:10.1038/srep24784, 2016.

*The authors thank the reviewer for suggesting additional literature. These references provide useful information for updating the introduction section of the manuscript.*

---

## Author Comment (AC2) · 23 Aug 2019

**Author response to the comments of RC2 by dr. Andreas Heinemeyer.**

In the text below, the authors respond to the comments given by referee 2. The comments of the referee are given as plain text, while the authors response is given in *italic*.

This study is an interesting one. It tackles an important issue, our ability to model and predict how blanket bog peatlands evolved and (based on their current C balance) could behave under future climate change. It also highlights the limitation of peatlands to sequester carbon in the long-term: as the C balance of input vs decomposition decreases over time, the ability for net sequestration is limited. This issue is often overlooked and needs to be highlighted

**Comments**

The authors could have added a few more modelling studies which already highlighted this issue (i.e. Heinemeyer et al., 2010; Frolking et al., 2010) – plenty of peatland development models show an asymptotic C accumulation over time, leveling off during the past few thousand years. However, this depends on the peatland conditions (climate and topography). A bit more context in the introduction and discussion would be beneficial.

*It was not the intention of the authors to give a full overview of the existing long-term peatland models and their advantages and disadvantages. Therefore, we referred to articles were this overview is given. However, we agree that this results in a short and maybe incomplete introduction to the literature on peatland modelling which might leave the reader with some unanswered questions.*

*As a result, we will make some changes to the manuscript to elaborate on the available studies dealing with the long-term modelling of (blanket) peatlands and give a short overview of the existing models. For a detailed description of the models however, we will refer to other articles to avoid a long and heavy manuscript.*

*The asymptotic behaviour of the carbon accumulation curve at millennial timescales is indeed an important result of this article. We will add a paragraph to the discussion to elaborate on the long-term carbon storage potential of peatlands and compare it to other studies as mentioned by the referee.*

Although the model is interesting there are a few points I raise (also see below). One important, related aspect is the recreated past climate. I do not think that the anomalies for temperature or rainfall are capturing past changes. Just think about the little ice age/medieval warm period etc. Maybe consider Heinemeyer et al,, 2010 and Morris et al., 2015 for some UK specific reconstructions. 8 mm difference in monthly rainfall seem somewhat meaningless. Also, the initial warming impact and wetting at the onset of the warming about 12k ya should be more pronounced. I suggest you consider some specific literature (although I acknowledge that actually data on this is not that easy to come by). I think the issue is mainly in using a large scale model output – actually differences are lost (same as when averaging current climate over large grid scales). I suggest you discuss this limitation in the light of the above concerns and publications.

*The authors agree that the climate reconstructions used in this article have their disadvantages, but quantitative climate data spanning the entire Holocene period are difficult to come by. In this study we use a pollen-based climate reconstruction which provides mean annual temperature and mean monthly precipitation anomalies for the past 12 000 years at a 1° x 1° spatial resolution and a 500-year temporal resolution (Mauri et al., 2015). Note that the UK specific reconstructions by Morris et al 2015 are also largely based on continental-scale gridded climate reconstructions such as those by Mauri et al 2015 which we used. UK specific reconstructions used by Heinemeyer et al 2010 are based on palaeoclimate records from the US and China and updated with present-day UK climate data. Hence, the UK specific reconstructions the referee is referring to are also not relying on UK-based palaeoclimate data. Furthermore, the Mauri dataset was rescaled to a 50m resolution gridded dataset by taking into account local orographic effects using meteorological data from 7 stations in the vicinity of the study area. Local information on climate is thus also used to finetune the palaeoclimate records and to increase the spatial variability that is not present in the continental-scale datasets. At present, we have no indications that one climate reconstruction is better than the other.*

*In the revised manuscript, we will add a few lines on the possible limitations and drawbacks of using these continental-scale climate reconstructions. Additionally, we will compare the used climate reconstruction with alternative sources as listed by the referee to contextualise the dataset used in this study.*

Moreover, the lack of a link between runoff and erosion within a hill slope context seems very odd to me. I suggest you consider this but maybe I have missed something in the methods. I just cannot find a link – which is crucial to allow slope C accumulation (which is not only decomposition [water table x temperature] driven). But your calibration will lead to overcompensation because of an important C flux process missing in your overall C budget.

*The erosion of peat is indeed not included in the model. This does not mean that the phenomenon of peatland erosion is not present in the study area. At multiple locations, erosion features can be detected on the hillslopes. However, we decided not to include this process for a couple of reasons. Firstly, the erosional features observed within the study area cover a range of processes including sheet erosion,*

*gullies, shallow mass movements, etc. This would complicate the model in correctly representing these different processes in a correct way. Secondly, the erosion of peatlands is a complex process with an entire body of literature dealing with this issue. The erosion equation as used by Heinemeyer et al 2010 predicts erosion of TOC in function of runoff depth and water table depth. This is a simple approach that neglects the variety in erosion processes observed. As we know from wealth of studies on erosion of mineral soils, erosion equations differ widely for sheet and rill erosion compared to gully erosion and various types of mass movement. At present, and to the authors knowledge, no comprehensive peat erosion model exists that can cover all peat erosion processes and that is easy to implement in a long-term peatland growth model. As a result, we chose not to implement this process in the model since it would increase model complexity, increase the calculation time and a proper calibration would be nearly impossibly due to the large number of parameters that need to be calibrated (in case all types of erosion processes would be simulated). Several erosion and hydrological modelling studies have shown that reducing model complexity is needed to reduce model uncertainties (e.g. Rompaey, A. J. V., & Govers, G. (2002). Data quality and model complexity for regional scale soil erosion prediction. International Journal of Geographical Information Science, 16(7), 663-680 or Jetten, V., Govers, G., & Hessel, R. (2003). Erosion models: quality of spatial predictions. Hydrological processes, 17(5), 887-900.).*

*We are aware that as a consequence, the calibrated decomposition rate as found during the model calibration procedure will be higher because it will encompass other processes (such as erosion) which remove peat mass from a certain location. We will add a paragraph to the manuscript to clarify this issue and the resulting effect on the calibration process and model output.*

This previous point also relates to the possible issue that underlying bedrock slope might not relate directly to peat surface slope (depressions) – another reviewer already made a comment on this. Would be nice to see a comparison in this respect.

*This comment was discussed in the answer to reviewer 1. The field data indicate a clear relationship between the bedrock slope and the peat thickness. Additionally, the relationship between the surface slope and the bedrock slope shows to be consistent with most gridpoints clustering close to the 1:1 line (see figure below). This scatterplot indicates that the observed range in slope values strongly exceeds the possible differences between the bedrock slope and the surface slope at a certain location. Hence, it demonstrates that the use of bedrock slope in the model (and thus, not adjusting the topographic slope when peat accumulates) is not likely to lead to a bias in modelled peat thicknesses.*

[Figure]

*Figure 1: Scatterplot of the surface slope and bedrock slope (in percent) for all coring locations.*

Very interesting to see such a high level of heather domination over time. Also the hill slope data are if general interest. Are those data to be made available? I suggest a section and/or doi for the data.

*The authors plan to make the hillslope data available by adding the data to the manuscript as supplementary material (see also our response to referee 1).*

*The raw data of some of the pollen cores used in the REVEALS analysis can be found in the open-source European Pollen Database (EPD) (Birks, 1969 & Huntley, 1994). The authors are planning to add the pollen data from Hunter, 2016 to the EPD as well. For the three remaining pollen cores used in the analysis, permission was granted by the original author (Paterson, 2011).*

Line 129: The underlying bedrock is impermeable (drainage). In most cases it is, and that will likely affect water tables during dry periods. They do have hydraulic conductivity and porosity... but in line 147 you mention that the mineral layer has been assigned a hydraulic conductivity. Explain and be consistent throughout the manuscript.

*All simulations start with a situation where there is an impermeable bedrock covered by a layer of mineral material, representing the glacial till deposits, which does have a porosity and hydraulic conductivity value. Over time, when organic matter keeps accumulating in the mineral layer and a threshold is exceeded, peat starts to form on top of this mineral layer. This means that at the start of the simulation, all gridpoints have two layers (bedrock and a mineral layer) and that throughout the simulation, some gridpoints can develop a third layer (peat) if the right conditions are available.*

*The authors will make sure that the paragraphs dealing with the model domain and the subsurface representation in the model are consistent in terminology to avoid confusion and possible misinterpretations.*

Line 143: The hydrology model just assumes that the water table depth is 'always' near the surface. This is not true. I bet in 2018 the water tables even in deeper and normally wet Scottish blanket bog went down to 30 cm or even more.

*This is not entirely the case. Indeed, the difference in saturated hydraulic conductivity between the acrotelm and catotelm within the peat profile will affect the water table behaviour but does not necessarily mean that the water table is located at this boundary. Some locations will be almost fully water saturated for most of the time while others will have a water table which is located much lower in the peat profile. For example, for the year 2010, which is the end date for the simulations, the calibrated thickness of the acrotelm is 10 centimetres while the mean water table depth is 22.3 centimetres below the surface. The water table dynamics are influenced by the difference in hydraulic conductivity, but also by the hillslope topography, which indicates the importance of 2D-simulations of blanket peatlands.*

Line 196: Litter input and quality (NPP) are affected by both, temperature and precipitation (Leith's equation). It is a bit odd to use the Moor House equation, which is purely temperature based. Maybe provide a critical assessment of doing so. Particularly the role of trees in peat depend on the water table depth (trees evaporate a lot and can cause shrinkage and decomposition of peat – but the understanding about this is still somewhat limited). Maybe consider recent publications on Scottish afforested bog by Sloan et al (with the late Richard Payne as his supervisor).

*The model does indeed calculate NPP based on an empirical relationship between NPP and the mean annual temperature using the Moor House dataset. There are a couple of reasons for this. Firstly, the equation of Lieth/Miami model uses both precipitation and temperature to calculate the NPP separately, using the minimum value of both for a given situation. Under the climatic conditions of the study area however, precipitation is not the limiting factor for the NPP, which will result in a NPP calculation which is determined by the mean annual temperature when using Lieth's equation. Secondly, the data from Moor*

*House indicate that for a given temperature, the NPP is lower than would be predicted based on Lieth's equation given a temperature range similar to the range observed within the study area. Since the data from Moor House come from a relatively similar environment to the study area, in contrast with the Miami model which was designed for a wide range of environments, we decided to work with an empirical equation based on the field data of Moor House. Indeed a similar result can be obtained using the approach in the MILLENNIA model with a conversion factor for the NPP calculation.*

*We agree that calculating a good NPP value based on the limited data available is difficult and that different approaches or equations can be used. We will add a few lines to the manuscript discussing in more detail why we chose this approach and to indicate the possible advantages and disadvantages of such an approach in comparison with the methods used in other peatland models.*

Line 148: Snow evaporation during frosty and dry periods can be important (see Carroll et al., 2015 model description).

*This process is indeed not included in the model. The authors are aware that some processes are not represented in detail in the model (such as snow sublimation, but also erosion  - see above). This is done to keep the model relatively simple and to reduce the calculation time necessary for model runs. The authors are of the opinion that the process of snow sublimation has a relatively limited impact on the total hillslope hydrology and as a result, the process is not explicitly represented in the model.*

Line 168: AET is also dependent on rooting depth in relation to the water table depth (see Carroll et al., 2015 model description).

*The representation of the AET within the model is indeed relatively simple. In contrast to more detailed models such as MILLENNIA or the HPM, we do not use plant functional types in our vegetation calculations. As a result, we used a more simplified representation of the water table dependence of the AET as described by Hilbert et al. 2000. There is no doubt that other existing peatland model better represent certain peatland processes. However, it is the intention of this study to come up with a relatively simple model which allows to be applied at a larger spatial scale. As a result, some processes are excluded from the model or have a simple representation. See also our discussion regarding the erosion processes not included in the model.*

Line 207: It seems a bit odd to have a 2 phase Q10 which will cause a jumping up and down in steps (why not have a continuous change in Q10 – simple function?). Also, the Q10 question is very questionable

indeed – apparent vs intrinsic & short-term vs long-term & with roots vs without roots - but that is a complex issue. It would be good to also provide references as to those two Q10 values.

*The use of a double q10-value is based on the review article of Chapman and Thurlow (1998) on peat respiration at low temperatures, which indicates that at lower temperatures (below 5°C), the q10-value is higher due to the fact that a temperature increase within this range expands the range of soil taxa, actively decomposing organic matter. The choice of q10-value is indeed not straightforward and a range of values is available. We will add a paragraph to the manuscript to explain the choice for two q10-values and to place the chosen values within the range mentioned in the literature to provide more background information.*

Please add a bit more information on what kind of model you developed/tested (empirical/process) "spatially-explicit hill slope model".

*We will provide some more details to clarify the kind of model presented in this article.*

Please add more information to what this means "peatland architecture was reconstructed". Currently this is unclear (did you take core samples/use a pollen and/or peat depth database?). Also, is 'assessed' a better word?

*The reconstruction consists of soil corings along hillslope transect, peat samples and seven pollen cores. We will clarify this in the manuscript.*

Please also add more information on what kind of climate data you used "climate reconstructions" e.g. mean annual precipitation/temperature … . The above points will help the reader grasp the complexity of what you did and put it into context for specific interests.

*In this article, we use the reconstructed mean annual temperature and mean monthly precipitation. We will make a few changes to the paragraph on the climate reconstruction to discuss in more detail the climate reconstructions used in this article.*

There are a lot of Gallego-Sala references when talking about basics of blanket bog. Is this the first time those explanations are made? I would have thought that Lindsay would have done this earlier and others before him. It is always good to go back to the original people.

*We will check the references and make sure we refer to the original article.*

I think it would be nice to mention the MILLENNIA (Heinemeyer et al., 2010) peatland model – coming out at the same time as HPM but for a UK context (so I suggest it is important in this context). This is also relevant when considering hill slope (see topography effects on temperature and hydrology, especially runoff and erosion - as in your study also using Garnett peat depth data for a validation at Moor House) and root C inputs. However, a valid point of criticism would be that it is a 2D model – so you are trying to do something better, which is great. However, you might also want to discuss what the disadvantages are of your simpler 2 depth model (root C input vs water table depth and porosity over depth affecting oxygen availability and thus decomposition – see Carroll et al. 2015 using the MILLENNIA model).

*There is no doubt that more detailed peatland models exist, such as the MILLENNIA model, Digibog, the Holocene Peatland Model, etc., which are better designed to capture specific processes such as the difference between above and below ground biomass or more gradual changes in peat properties throughout the peat profile. However, the idea behind this paper is to demonstrate that when a model is constructed with a simplified representation of the processes at play in a peat profile but incorporating an additional spatial dimension to study hillslope hydrology, we are able to reconstruct long-term peatland dynamics at the landscape-scale, while point-by-point comparison is more difficult. This is discussed throughout the manuscript, but we will strengthen this message in the discussion to indicate more clearly the difference between the existing detailed models such as MILLENNIA, Digibog, HPM, … and the model presented in this paper, highlighting possible advantages and disadvantages of this relatively simple peatland model in studying blanket peatlands.*

Finally, the empirical C input relationship with only temperature seems questionable when NPP is clearly dependent on both, temperature and precipitation – potential evapotranspiration functions are commonly used as in the MILLENNIA model, Durham Carbon Model by Worrall et al. etc.).

*This is discussed in detail in a comment above.*

Line 99: Maybe better to use 'assessed' instead of reconstructed. L100 Soil cores were 'taken' along … L104 give details of the sections (what depth ranges were assessed – BD changes with depth).

*This will be changed in the manuscript.*

Line 131: Please add more information regarding "Boussinesq equation for the simulation of the hillslope hydrology".

*We will include a few lines to elaborate on the Boussinesq-equation since the water table behaviour is an important part of the peatland model.*

Peat Initiation I like that there is a threshold for 'peat' based on C accumulation in the top mineral layer! Nice work. However, I suggest that a bit more information is needed again for the definition of the threshold (%Corg?).

*This is indeed based on a threshold value. The mineral layer accumulates organic matter until the upper horizon of the mineral substrate contains the amount equivalent to the organic carbon mass in a peat layer of 10 centimetres (which is also the threshold we used in the field for the definition of peat). We will clarify this in the manuscript.*

Model Calibration I think it would be nice to see how all the parameters changed from default to calibrated. Possibly also how they compare to other publications (if applicable; e.g. HPD).

*This is indeed an interesting comparison and a paragraph will be added to the discussion to locate the calibrated parameters within the range of values mentioned in the literature.*

Results I think reporting dry bulk density values is best in g/cm3 – most models and field measurements in peatlands will show those units (unless I am mistaken).

*This unit is indeed more in line with other studies and will be changed in the manuscript.*